# Stability Guarantees for Feature Attributions with Multiplicative Smoothing

**Anton Xue**   **Rajeev Alur**   **Eric Wong**
Department of Computer and Information Science
University of Pennsylvania
Philadelphia, PA 19104
{antonxue,alur,exwong}@seas.upenn.edu

## Abstract

Explanation methods for machine learning models tend not to provide any formal guarantees and may not reflect the underlying decision-making process. In this work, we analyze stability as a property for reliable feature attribution methods. We prove that relaxed variants of stability are guaranteed if the model is sufficiently Lipschitz with respect to the masking of features. We develop a smoothing method called Multiplicative Smoothing (MuS) to achieve such a model. We show that MuS overcomes the theoretical limitations of standard smoothing techniques and can be integrated with any classifier and feature attribution method. We evaluate MuS on vision and language models with various feature attribution methods, such as LIME and SHAP, and demonstrate that MuS endows feature attributions with non-trivial stability guarantees.

## 1 Introduction

Modern machine learning models are incredibly powerful at challenging prediction tasks but notoriously black-box in their decision-making. One can therefore achieve impressive performance without fully understanding *why*. In settings like medical diagnosis [1, 2] and legal analysis [3, 4], where accurate and well-justified decisions are important, such power without proof is insufficient. In order to fully wield the power of such models while ensuring reliability and trust, a user needs accurate and insightful *explanations* of model behavior.

One popular family of explanation methods is *feature attributions* [5, 6, 7, 8]. Given a model and input, a feature attribution method generates a score for each input feature that denotes its importance to the overall prediction. For instance, consider Figure 1, in which the Vision Transformer [9] classifier predicts the full image (left) as "Goldfish". We then use a feature attribution method like SHAP [7] to score each feature and select the top-25%, for which the masked image (middle) is consistently predicted as "Goldfish". However, adding a single patch of features (right) alters the prediction confidence so much that it now yields "Axolotl". This suggests that the explanation is brittle [10], as small changes easily cause it to induce some other class. In this paper, we study how to overcome such behavior by analyzing the *stability* of an explanation: we consider an explanation to be stable if, once the explanatory features are included, the addition of more features does not change the prediction.

Stability implies that the selected features are enough to explain the prediction [12, 13, 14] and that this selection maintains strong explanatory power even in the presence of additional information [10, 15]. Similar properties are studied in literature and identified as useful for interpretability [16], and we emphasize that our main focus is on analyzing and achieving provable guarantees. Stability guarantees, in particular, are useful as they allow one to predict how model behavior varies with the explanation. Given a stable explanation, one can include more features, e.g., adding context, while

37th Conference on Neural Information Processing Systems (NeurIPS 2023).

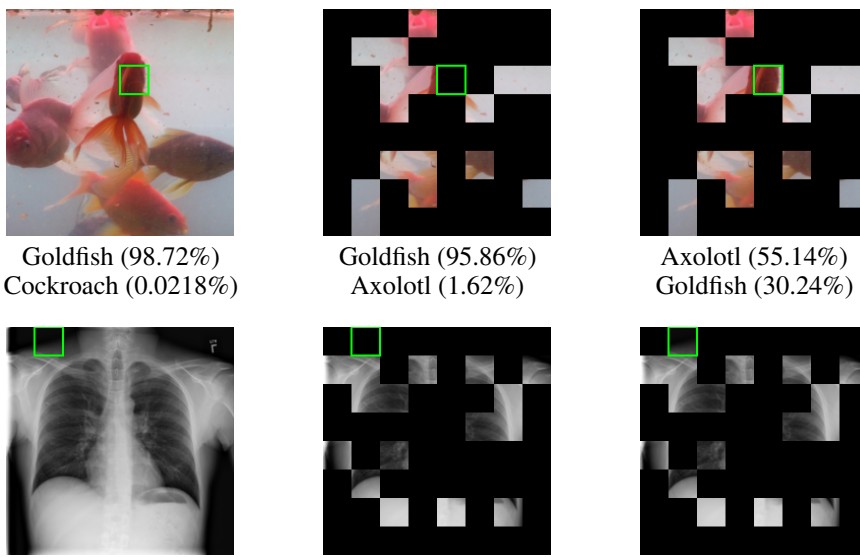

| Goldfish (98.72%) | Goldfish (95.86%) | Axolotl (55.14%) |
| Cockroach (0.0218%) | Axolotl (1.62%) | Goldfish (30.24%) |

| Pneumonia (25.50%) | Pneumonia (29.56%) | Pneumonia (50.05%) |

Figure 1: (Top) Classification by Vision Transformer [9]. (Bottom) Pneumonia detection from an X-ray image by DenseNet-Res224 from TorchXRayVision [11]. Both are $224 \times 224$ pixel images whose attributions are derived from SHAP [7] with top-25% selection. A single $28 \times 28$ pixel patch of difference between the two attributions (marked **green**) significantly affects prediction confidence.

maintaining confidence in the consistency of the underlying explanatory power. Crucially, we observe that such guarantees only make sense when jointly considering the model and explanation method: the explanation method necessarily depends on the model to yield an explanation, and stability is then evaluated with respect to the model.

Thus far, existing works on feature attributions with formal guarantees face computational tractability and explanatory utility challenges. While some methods take an axiomatic approach [8, 17], others use metrics that appear reasonable but may not reliably reflect useful model behavior, a common and known limitation [18]. Such explanations have been criticized as a plausible guess at best, and completely misleading [19] at worst.

In this paper, we study how to construct explainable models with provable stability guarantees. We jointly consider the classification model and explanation method and present a formalization for studying such properties that we call *explainable models*. We focus on *binary feature attributions* [20] wherein each feature is either marked as explanatory (1) or not explanatory (0). We present a method to solve this problem, inspired by techniques from adversarial robustness, particularly randomized smoothing [21, 22]. Our method can take *any* off-the-shelf classifier and feature attribution method to efficiently yield an explainable model that satisfies provable stability guarantees. In summary, our contributions are as follows:

- We formalize stability as a key property for binary feature attributions and study this in the framework of explainable models. We prove that relaxed variants of stability are guaranteed if the model is sufficiently Lipschitz with respect to the masking of features.

- To achieve the sufficient Lipschitz condition, we develop a smoothing method called Multiplicative Smoothing (MuS). We show that MuS achieves strong smoothness conditions, overcomes key theoretical and practical limitations of standard smoothing techniques, and can be integrated with any classifier and feature attribution method.

- We evaluate MuS on vision and language models along with different feature attribution methods. We demonstrate that MuS-smoothed explainable models achieve strong stability guarantees at a small cost to accuracy.

## 2 Overview

We observe that formal guarantees for explanations must take into account both the model and explanation method, and for this, we present in Section 2.1 a pairing that we call *explainable models*. This formulation allows us to describe the desired stability properties in Section 2.2. We show in Section 2.3 that a classifier with sufficient Lipschitz smoothness with respect to feature masking allows us to yield provable guarantees of stability. Finally, in Section 2.4, we show how to adapt existing feature attribution methods into our explainable model framework.

### 2.1 Explainable Models

We first present explainable models as a formalism for rigorously studying explanations. Let $\mathcal{X} = \mathbb{R}^n$ be the space of inputs, a classifier $f : \mathcal{X} \to [0,1]^m$ maps inputs $x \in \mathcal{X}$ to $m$ class probabilities that sum to 1, where the class of $f(x) \in [0,1]^m$ is taken to be the largest coordinate. Similarly, an explanation method $\varphi : \mathcal{X} \to \{0,1\}^n$ maps an input $x \in \mathcal{X}$ to an explanation $\varphi(x) \in \{0,1\}^n$ that indicates which features are considered explanatory for the prediction $f(x)$. In particular, we may pick and adapt $\varphi$ from among a selection of existing feature attribution methods like LIME [6], SHAP [7], and many others [5, 8, 23, 24, 25], wherein $\varphi$ may be thought of as a top-$k$ feature selector. Note that the selection of input features necessarily depends on the explanation method executing or analyzing the model, and so it makes sense to jointly study the model and explanation method: given a classifier $f$ and explanation method $\varphi$, we call the pairing $\langle f, \varphi \rangle$ an *explainable model*. Given some $x \in \mathcal{X}$, the explainable model $\langle f, \varphi \rangle$ maps $x$ to both a prediction and explanation. We show this in Figure 2, where $\langle f, \varphi \rangle(x) \in [0,1]^m \times \{0,1\}^n$ pairs the class probabilities and the feature attribution.

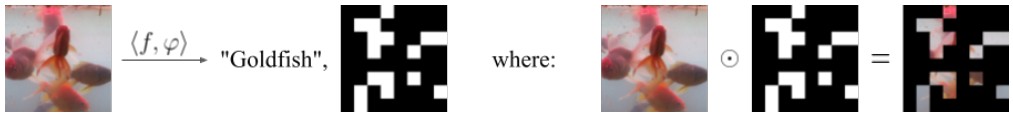

Figure 2: An explainable model $\langle f, \varphi \rangle$ outputs both a classification and a feature attribution. The feature attribution is a binary-valued mask (white 1, black 0) that can be applied over the original input. Here $f$ is Vision Transformer [9] and $\varphi$ is SHAP [7] with top-25% feature selection.

For an input $x \in \mathcal{X}$, we will evaluate the quality of the binary feature attribution $\varphi(x)$ through its masking on $x$. That is, we will study the behavior of $f$ on the masked input $x \odot \varphi(x) \in \mathcal{X}$, where $\odot$ is the element-wise vector product. To do this, we define a notion of *prediction equivalence*: for two $x, x' \in \mathcal{X}$, we write $f(x) \cong f(x')$ to mean that $f(x)$ and $f(x')$ yield the same class. This allows us to formalize the intuition that an explanation $\varphi(x)$ should recover the prediction of $x$ under $f$.

**Definition 2.1.** *The explainable model $\langle f, \varphi \rangle$ is consistent at $x$ if $f(x) \cong f(x \odot \varphi(x))$.*

Evaluating $f$ on $x \odot \varphi(x)$ this way lets us apply the model as-is and therefore avoids the challenge of constructing a surrogate model that is accurate to the original [26]. Moreover, this approach is reasonable, especially in domains like vision — where one intuitively expects that a masked image retaining only the important features should induce the intended prediction. Indeed, architectures like Vision Transformer [9] can maintain high accuracy with only a fraction of the image present [27].

Particularly, we would like for $\langle f, \varphi \rangle$ to generate explanations that are stable and concise (i.e. sparse). The former is our central guarantee and is ensured through smoothing. The latter implies that $\varphi(x)$ has few ones entries, and is a desirable property since a good explanation should not contain too much redundant information. However, sparsity is a more difficult property to enforce, as this is contingent on the model having high accuracy with respect to heavily masked inputs. For sparsity, we present a simple heuristic in Section 2.4 and evaluate its effectiveness in Section 4.

### 2.2 Stability Properties of Explainable Models

Given an explainable model $\langle f, \varphi \rangle$ and some $x \in \mathcal{X}$, stability means that the prediction does not change even if one adds more explanatory features to $\varphi(x)$. For instance, the model-explanation pair in Figure 1 is *not* stable, as the inclusion of a single feature group (patch) changes the prediction. To formalize this notion of stability, we first introduce a partial ordering: for $\alpha, \alpha' \in \{0,1\}^n$, we write $\alpha \succeq \alpha'$ iff $\alpha_i \geq \alpha'_i$ for all $i = 1, \ldots, n$. That is, $\alpha \succeq \alpha'$ iff $\alpha$ includes all the features selected by $\alpha'$.

**Definition 2.2.** *The explainable model $\langle f, \varphi \rangle$ is stable at $x$ if $f(x \odot \alpha) \cong f(x \odot \varphi(x))$ for all $\alpha \succeq \varphi(x)$.*

Note that the constant explanation $\varphi(x) = \mathbf{1}$, the vector of ones, makes $\langle f, \varphi \rangle$ trivially stable at every $x \in \mathcal{X}$, though this is not a concise explanation. Additionally, stability at $x$ implies consistency at $x$.

Unfortunately, stability is a difficult property to enforce in general, as it requires that $f$ satisfy a monotone-like behavior with respect to feature inclusion — which is especially challenging for complex models like neural networks. Checking stability without additional assumptions on $f$ is also hard: if $k = \|\varphi(x)\|_1$ is the number of ones in $\varphi(x)$, then there are $2^{n-k}$ possible $\alpha \succeq \varphi(x)$ to check. This large space of possible $\alpha \succeq \varphi(x)$ motivates us to examine instead *relaxations* of stability. We introduce lower and upper relaxations of stability below.

**Definition 2.3.** *The explainable model $\langle f, \varphi \rangle$ is incrementally stable at $x$ with radius $r$ if $f(x \odot \alpha) \cong f(x \odot \varphi(x))$ for all $\alpha \succeq \varphi(x)$ where $\|\alpha - \varphi(x)\|_1 \leq r$.*

Incremental stability is the lower relaxation since it considers the case where the mask $\alpha$ has only a few features more than $\varphi(x)$. For instance, if one can probably add up to $r$ features to a masked $x \odot \varphi(x)$ without altering the prediction, then $\langle f, \varphi \rangle$ would be incrementally stable at $x$ with radius $r$. We next introduce the upper relaxation that we call decremental stability.

**Definition 2.4.** *The explainable model $\langle f, \varphi \rangle$ is decrementally stable at $x$ with radius $r$ if $f(x \odot \alpha) \cong f(x \odot \varphi(x))$ for all $\alpha \succeq \varphi(x)$ where $\|\mathbf{1} - \alpha\|_1 \leq r$.*

Decremental stability is a subtractive form of stability, in contrast to the additive nature of incremental stability. Particularly, decremental stability considers the case where $\alpha$ has much more features than $\varphi(x)$. If one can provably remove up to $r$ non-explanatory features from the full $x$ without altering the prediction, then $\langle f, \varphi \rangle$ is decrementally stable at $x$ with radius $r$. Note also that decremental stability necessarily entails consistency of $\langle f, \varphi \rangle$, but for simplicity of definitions, we do not enforce this for incremental stability. Furthermore, observe that for a sufficiently large radius of $r = \lceil (n - \|\varphi(x)\|_1)/2 \rceil$, incremental and decremental stability together imply stability.

**Remark 2.5.** *Similar notions to the above have been proposed in the literature, and we refer to [16] for an extensive survey. In particular, for [16], consistency is akin to* preservation*, and stability is similar to* continuity*, except we are concerned with adding features. In this regard, incremental stability is most similar to* incremental addition *and decremental stability to* incremental deletion*.*

## 2.3 Lipschitz Smoothness Entails Stability Guarantees

If $f : \mathcal{X} \to [0, 1]^m$ is Lipschitz with respect to the masking of features, then we can guarantee relaxed stability properties for the explainable model $\langle f, \varphi \rangle$. In particular, we require for all $x \in \mathcal{X}$ that $f(x \odot \alpha)$ is Lipschitz with respect to the mask $\alpha \in \{0, 1\}^n$. This then allows us to establish our main result (Theorem 3.3), which we preview below in Remark 2.6.

**Remark 2.6** (Sketch of main result)**.** *Consider an explainable model $\langle f, \varphi \rangle$ where for all $x \in \mathcal{X}$ the function $g(x, \alpha) = f(x \odot \alpha)$ is $\lambda$-Lipschitz in $\alpha \in \{0, 1\}^n$ with respect to the $\ell^1$ norm. Then at any $x$, the radius of incremental stability $r_{\mathrm{inc}}$ and radius of decremental stability $r_{\mathrm{dec}}$ are respectively:*

$$r_{\mathrm{inc}} = \frac{g_A(x, \varphi(x)) - g_B(x, \varphi(x))}{2\lambda}, \qquad r_{\mathrm{dec}} = \frac{g_A(x, \mathbf{1}) - g_B(x, \mathbf{1})}{2\lambda},$$

*where $g_A - g_B$ is called the confidence gap, with $g_A, g_B$ the top-two class probabilities:*

$$k^{\star} = \underset{1 \leq k \leq m}{\mathrm{argmax}}\, g_k(x, \alpha), \qquad g_A(x, \alpha) = g_{k^{\star}}(x, \alpha), \qquad g_B(x, \alpha) = \max_{i \neq k^{\star}} g_i(x, \alpha). \qquad (1)$$

Observe that Lipschitz smoothness is, in fact, a stronger assumption than necessary, as besides $\alpha \succeq \varphi(x)$, it also imposes guarantees on $\alpha \preceq \varphi(x)$. Nevertheless, Lipschitz smoothness is one of the few classes of properties that can be guaranteed and analyzed at scale on arbitrary models [22, 28].

## 2.4 Adapting Existing Feature Attribution Methods

Most existing feature attribution methods assign a real-valued score to feature importance, rather than a binary value. We therefore need to convert this to a binary-valued method for use with a

stable explainable model. Let $\psi : \mathcal{X} \to \mathbb{R}^n$ be such a continuous-valued method like LIME [6] or SHAP [7], and fix some desired incremental stability radius $r_{\text{inc}}$ and decremental stability radius $r_{\text{dec}}$. Given some $x \in \mathcal{X}$ a simple construction for binary $\varphi(x) \in \{0, 1\}^n$ is described next.

**Remark 2.7** (Iterative construction of $\varphi(x)$). *Consider any $x \in \mathcal{X}$ and let $\rho$ be an index ordering on $\psi(x)$ from high-to-low (i.e. most probable class first). Initialize $\alpha = \mathbf{0}$, and for each $i \in \rho$: assign $\alpha_i \leftarrow 1$ then check whether $\langle f, \varphi : x \mapsto \alpha \rangle$ is now consistent, incrementally stable with radius $r_{\text{inc}}$, and decrementally stable with radius $r_{\text{dec}}$. If so, terminate with $\varphi(x) = \alpha$, and continue otherwise.*

Note that the above method of constructing $\varphi(x)$ does not impose sparsity guarantees in the way that we may guarantee stability through Lipschitz smoothness. Instead, the ordering from a continuous-valued $\psi(x)$ serves as a greedy heuristic for constructing $\varphi(x)$. We show in Section 4 that some feature attributions (e.g., SHAP [7]) tend to yield sparser selections on average compared to others (e.g., Vanilla Gradient Saliency [5]).

## 3   Multiplicative Smoothing for Lipschitz Constants

In this section, we present our main technical contribution of Multiplicative Smoothing (MuS). The goal is to transform an arbitrary base classifier $h : \mathcal{X} \to [0, 1]^m$ into a smoothed classifier $f : \mathcal{X} \to [0, 1]^m$ that is Lipschitz with respect to the masking of features. This then allows one to couple $f$ with an explanation method $\varphi$ in order to form an explainable model $\langle f, \varphi \rangle$ with provable stability guarantees.

We give an overview of our MuS in Section 3.1, where we illustrate that a principal motivation for its development is because standard smoothing techniques may violate a property that we call *masking equivalence*. We present the Lipschitz constant of the smoothed classifier $f$ in Section 3.2 and show how this is used to certify stability. Finally, we give an efficient computation of MuS in Section 3.3, allowing us to exactly evaluate $f$ at a low sample complexity.

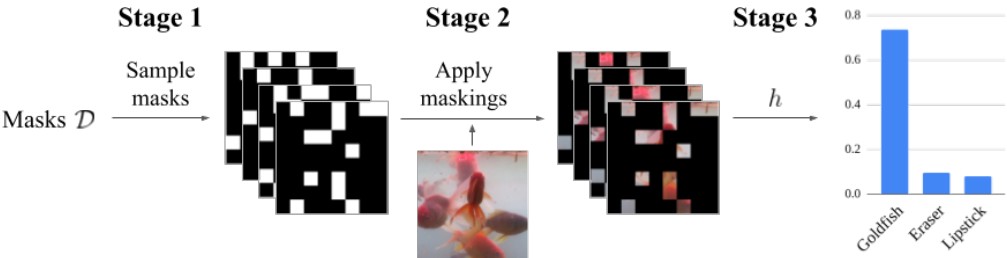

Figure 3: Evaluating $f(x)$ is done in three stages. **(Stage 1)** Generate $N$ samples of binary masks $s^{(1)}, \ldots, s^{(N)} \in \{0, 1\}^n$, where each coordinate is Bernoulli with parameter $\lambda$ (here $\lambda = 1/4$). **(Stage 2)** Apply each mask on the input to yield $x \odot s^{(i)}$ for $i = 1, \ldots, N$. **(Stage 3)** Average over $h(x \odot s^{(i)})$ to compute $f(x)$, and note that the predicted class is given by a weighted average.

### 3.1   Technical Overview of MuS

Our key insight is that randomly dropping (i.e., zeroing) features will attain the desired smoothness. In particular, we uniformly drop features with probability $1 - \lambda$ by sampling binary masks $s \in \{0, 1\}^n$ from some distribution $\mathcal{D}$ where each coordinate is distributed as $\Pr[s_i = 1] = \lambda$. Then define $f$ as:

$$f(x) = \mathop{\mathbb{E}}_{s \sim \mathcal{D}} h(x \odot s), \qquad \text{such that } s_i \sim \mathcal{B}(\lambda) \text{ for } i = 1, \ldots, n \tag{2}$$

where $\mathcal{B}(\lambda)$ is the Beronulli distribution with parameter $\lambda \in [0, 1]$. We give an overview of evaluating $f(x)$ in Figure 3. Importantly, our main results on smoothness (Theorem 3.2) and stability (Theorem 3.3) hold provided each coordinate of $\mathcal{D}$ is marginally Bernoulli with parameter $\lambda$, and so we avoid fixing a particular choice for now. However, it will be easy to intuit the exposition with $\mathcal{D} = \mathcal{B}^n(\lambda)$, the coordinate-wise i.i.d. Bernoulli distribution with parameter $\lambda$.

We can equivalently parametrize $f$ using the mapping $g(x, \alpha) = f(x \odot \alpha)$, where it follows that:

$$g(x, \alpha) = \mathop{\mathbb{E}}_{s \sim \mathcal{D}} h(x \odot \tilde{\alpha}), \qquad \tilde{\alpha} = \alpha \odot s. \tag{3}$$

Note that one could have alternatively first defined $g$ and then $f$ due to the identity $g(x, \mathbf{1}) = f(x)$. We require that the relationship between $f$ and $g$ follows an identity that we call *masking equivalence*:

$$g(x \odot \alpha, \mathbf{1}) = f(x \odot \alpha) = g(x, \alpha), \qquad \text{for all } x \in \mathcal{X} \text{ and } \alpha \in \{0,1\}^n. \tag{4}$$

This follows by the definition of $g$, and the relevance to stability is this: if masking equivalence holds, then we can rewrite stability properties involving $f$ in terms of $g$'s second parameter as follows:

$$f(x \odot \alpha) = g(x, \alpha) \cong g(x, \varphi(x)) = f(x \odot \varphi(x)), \qquad \text{for all } \alpha \succeq \varphi(x), \quad \text{(c.f. Definition 2.2)}$$

where incremental and decremental stability may be analogously defined. This translation is useful, as we will prove that $g$ is $\lambda$-Lipschitz in its second parameter (Theorem 3.2), which then allows us to establish the desired stability properties (Theorem 3.3).

Since many choices are valid, we have not given an explicit construction for $\mathcal{D}$. Rather, so long as each coordinate of $s \sim \mathcal{D}$ obeys $s_i \sim \mathcal{B}(\lambda)$ then the Lipschitz properties for $g$ follow. The implication here is that although simple distributions like $\mathcal{B}^n(\lambda)$ suffice for $\mathcal{D}$, they may not be sample efficient. We show in Section 3.3 how to exploit a structured statistical dependence in order to reduce the sample complexity of computing MuS.

Importantly, we are motivated to develop MuS because standard smoothing techniques, namely additive smoothing [21, 22], may fail to satisfy masking equivalence. Additive smoothing is by far the most popular smoothing technique and differs from our scheme (3) in how noise is applied, where let $\mathcal{D}_{\mathrm{add}}$ and $\mathcal{D}_{\mathrm{mult}}$ be any two distributions on $\mathbb{R}^n$:

$$g(x, \alpha) = \mathop{\mathbb{E}}_{s \sim \mathcal{D}} h(x \odot \tilde{\alpha}), \qquad \tilde{\alpha} = \begin{cases} \alpha + s, & s \sim \mathcal{D}_{\mathrm{add}}, & \text{additive noise} \\ \alpha \odot s, & s \sim \mathcal{D}_{\mathrm{mult}}, & \text{multiplicative noise} \end{cases}$$

Particularly, additive smoothing has counterexamples for masking equivalence.

**Proposition 3.1.** *There exists $h : \mathcal{X} \to [0, 1]$ and distribution $\mathcal{D}$, where for*

$$g^+(x, \alpha) = \mathop{\mathbb{E}}_{s \sim \mathcal{D}} h(x \odot \tilde{\alpha}), \qquad \tilde{\alpha} = \alpha + s,$$

*we have $g^+(x, \alpha) \neq g^+(x \odot \alpha, \mathbf{1})$ for some $x \in \mathcal{X}$ and $\alpha \in \{0, 1\}^n$.*

*Proof.* Observe that it suffices to have $h, x, \alpha$ such that $h(x \odot (\alpha + s)) > h((x \odot \alpha) \odot (\mathbf{1} + s))$ for a non-empty set of $s \in \mathbb{R}^n$. Let $\mathcal{D}$ be a distribution on these $s$, then:

$$g^+(x, \alpha) = \mathop{\mathbb{E}}_{s \sim \mathcal{D}} h(x \odot (\alpha + s)) > \mathop{\mathbb{E}}_{s \sim \mathcal{D}} h((x \odot \alpha) \odot (\mathbf{1} + s)) = g^+(x \odot \alpha, \mathbf{1})$$

$\square$

Intuitively, this occurs because additive smoothing primarily applies noise by perturbing feature values, rather than completely masking them. As such, there might be "information leakage" when non-explanatory bits of $\alpha$ are changed into non-zero values. This then causes each sample of $h(x \odot \tilde{\alpha})$ within $g(x, \alpha)$ to observe more features of $x$ than it would have been able to otherwise.

## 3.2 Certifying Stability Properties with Lipschitz Classifiers

Our core technical result is in showing that $f$ as defined in (2) is Lipschitz to the masking of features. We present MuS in terms of $g$, where it is parametric with respect to the distribution $\mathcal{D}$: so long as $\mathcal{D}$ satisfies a coordinate-wise Bernoulli condition, then it is usable with MuS.

**Theorem 3.2** (MuS). *Let $\mathcal{D}$ be any distribution on $\{0,1\}^n$ where each coordinates of $s \sim \mathcal{D}$ is distributed as $s_i \sim \mathcal{B}(\lambda)$. Consider any $h : \mathcal{X} \to [0,1]$ and define $g : \mathcal{X} \times \{0,1\}^n \to [0,1]$ as*

$$g(x, \alpha) = \mathop{\mathbb{E}}_{s \sim \mathcal{D}} h(x \odot \tilde{\alpha}), \qquad \tilde{\alpha} = \alpha \odot s.$$

*Then the function $g(x, \cdot) : \{0,1\}^n \to [0,1]$ is $\lambda$-Lipschitz in the $\ell^1$ norm for all $x \in \mathcal{X}$.*

The strength of this result is in its weak assumptions. First, the theorem applies to any model $h$ and input $x \in \mathcal{X}$. It further suffices that each coordinate is distributed as $s_i \sim \mathcal{B}(\lambda)$, and we emphasize that statistical independence between different $s_i, s_j$ is *not assumed*. This allows us to construct

$\mathcal{D}$ with structured dependence in Section 3.3, such that we may exactly and efficiently evaluate $g(x, \alpha)$ at a sample complexity of $N \ll 2^n$. A low sample complexity is important for making MuS practically usable, as otherwise, one must settle for the expected value subject to probabilistic guarantees. For instance, simpler distributions like $\mathcal{B}^n(\lambda)$ do satisfy the requirements of Theorem 3.2 — but costs $2^n$ samples because of coordinate-wise independence. Whatever choice of $\mathcal{D}$, one can guarantee stability so long as $g$ is Lipschitz in its second argument.

**Theorem 3.3** (Stability). *Consider any* $h : \mathcal{X} \to [0, 1]^m$ *with coordinates* $h_1, \ldots, h_m$. *Fix* $\lambda \in [0, 1]$ *and let* $g_1, \ldots, g_m$ *be the respectively smoothed coordinates as in Theorem 3.2, using which we analogously define* $g : \mathcal{X} \times \{0, 1\}^n \to [0, 1]^m$. *Also define* $f(x) = g(x, \mathbf{1})$. *Then for any explanation method* $\varphi$ *and input* $x \in \mathcal{X}$, *the explainable model* $\langle f, \varphi \rangle$ *is incrementally stable with radius* $r_{\mathrm{inc}}$ *and decrementally stable with radius* $r_{\mathrm{dec}}$:

$$r_{\mathrm{inc}} = \frac{g_A(x, \varphi(x)) - g_B(x, \varphi(x))}{2\lambda}, \qquad r_{\mathrm{dec}} = \frac{g_A(x, \mathbf{1}) - g_B(x, \mathbf{1})}{2\lambda},$$

*where* $g_A - g_B$ *is the confidence gap as in* (1).

Note that it is only in the case where the radius $\geq 1$ that non-trivial stability guarantees exist. Because each $g_k$ has range in $[0, 1]$, this means that a Lipschitz constant of $\lambda \leq 1/2$ is necessary to attain at least one radius of stability. We present in Appendix A.2 some extensions to MuS that allow one to achieve higher coverage of features.

### 3.3 Exploiting Structured Dependency

We now present $\mathcal{L}_{qv}(\lambda)$, a distribution on $\{0, 1\}^n$ that allows for efficient and exact evaluation of a MuS-smoothed classifier. Our construction is an adaption of [28] from uniform to Bernoulli noise, where the primary insight is that one can parametrize $n$-dimensional noise using a single dimension via structured coordinate-wise dependence. In particular, we use a *seed vector* $v$, where with an integer *quantization parameter* $q > 1$ there will only exist $q$ distinct choices of $s \sim \mathcal{L}_{qv}(\lambda)$. All the while, we still enforce that any such $s$ is coordinate-wise Bernoulli with $s_i \sim \mathcal{B}(\lambda)$. Thus, for a sufficiently small quantization parameter (i.e., $q \ll 2^n$), we may tractably enumerate through all $q$ possible choices of $s$ and thereby evaluate a MuS-smoothed model with only $q$ samples.

**Proposition 3.4.** *Fix integer* $q > 1$ *and consider any vector* $v \in \{0, 1/q, \ldots, (q-1)/q\}^n$ *and scalar* $\lambda \in \{1/q, \ldots, q/q\}$. *Define* $s \sim \mathcal{L}_{qv}(\lambda)$ *to be a random vector in* $\{0, 1\}^n$ *with coordinates given by*

$$s_i = \mathbb{I}[t_i \leq \lambda], \qquad t_i = v_i + s_{\mathrm{base}} \mod 1, \qquad s_{\mathrm{base}} \sim \mathcal{U}(\{1/q, \ldots, q/q\}) - 1/(2q).$$

*Then there are* $q$ *distinct values of* $s$ *and each coordinate is distributed as* $s_i \sim \mathcal{B}(\lambda)$.

*Proof.* First, observe that each of the $q$ distinct values of $s_{\mathrm{base}}$ defines a unique value of $s$ since we have assumed $v$ and $\lambda$ to be fixed. Next, observe that each $t_i$ has $q$ unique values uniformly distributed as $t_i \sim \mathcal{U}(1/q, \ldots, q/q\}) - 1/(2q)$. Because $\lambda \in \{1/q, \ldots, q/q\}$ we therefore have $\Pr[t_i \leq \lambda] = \lambda$, which implies that $s_i \sim \mathcal{B}(\lambda)$. $\qquad \square$

The seed vector $v$ is the source of our structured coordinate-wise dependence, and the one-dimensional source of randomness $s_{\mathrm{base}}$ is used to generate the $n$-dimensional $s$. Such $s \sim \mathcal{L}_{qv}(\lambda)$ then satisfies the conditions for use in MuS (Theorem 3.2), and this noise allows for an exact evaluation of the smoothed classifier in $q$ samples. We have found $q = 64$ to be sufficient in practice and that values as low as $q = 16$ also yield good performance. We remark that one drawback is that one may get an unlucky seed $v$, but we have not yet observed this in our experiments.

## 4 Empirical Evaluations

We evaluate the quality of MuS on different classification models and explanation methods as they relate to stability guarantees. To that end, we perform the following experiments.

**(E1) How good are the stability guarantees?** A natural measure of quality for stability guarantees over a dataset exists: what radii are achieved, and at what frequency. We investigate how different combinations of models, explanation methods, and $\lambda$ affect this measure.

**(E2) What is the cost of smoothing?** To increase the radius of a provable stability guarantee, we must decrease the Lipschitz constant $\lambda$. However, as $\lambda$ decreases, more features are dropped during the smoothing process. This experiment investigates this stability-accuracy trade-off.

**(E3) Which explanation method is best?** We evaluate which existing feature attribution methods are amenable to strong stability guarantees. We examine LIME [6], SHAP [7], Vanilla Gradient Saliency (VGrad) [5], and Integrated Gradients (IGrad) [8], with focus on the size of the explanation.

**(Experimental Setup)** We use on two vision models (Vision Transformer [9] and ResNet50 [29]) and one language model (RoBERTa [30]). We use ImageNet1K [31] as our vision dataset and Tweet-Eval [32] sentiment analysis as our language dataset. We use *feature grouping* from Appendix A.2.1 on ImageNet1K to reduce the $3 \times 224 \times 224$ dimensional input into $n = 64$ superpixel patches. We report stability radii $r$ in terms of the fraction of features, i.e., $r/n$. In all our experiments, we use the quantized noise as in Section 3.3 with $q = 64$ unless specified otherwise. We refer to Appendix B for training details and the comprehensive experiments.

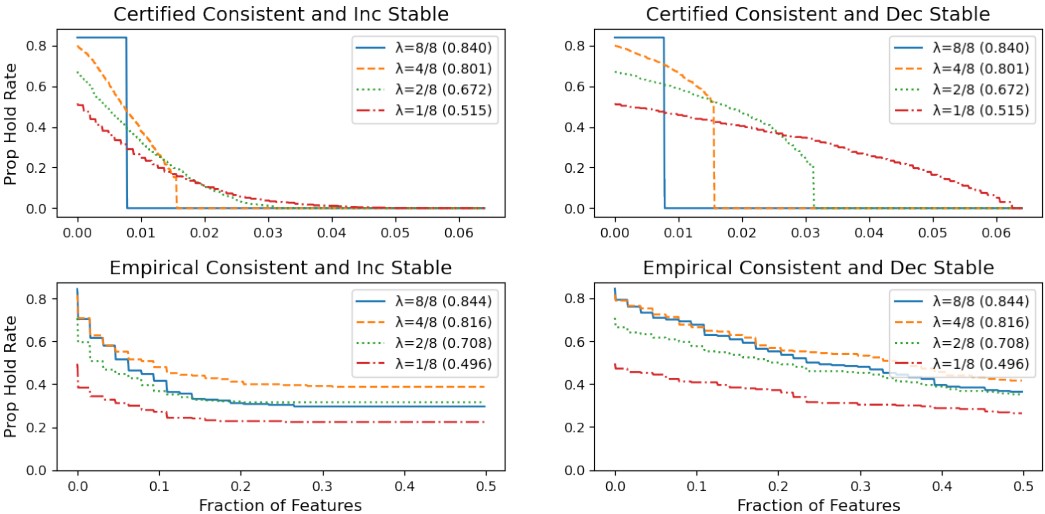

Figure 4: Rate of consistency and incremental (decremental) stability up to radius $r$ vs. fraction of feature coverage $r/n$. Left: certified $N_{\mathrm{cert}} = 2000$; Right: empirical $N_{\mathrm{emp}} = 250$ with $q = 16$.

## 4.1 (E1) Quality of Stability Guarantees

We study how much radius of consistent and incremental (resp. decremental) stability is achieved, and how often. We take an explainable model $\langle f, \varphi \rangle$ where $f$ is Vision Transformer and $\varphi$ is SHAP with top-25% feature selection. We plot the rate at which a property holds (e.g., consistent and incrementally stable with radius $r$) as a function of radius (expressed as a fraction of features $r/n$).

We show our results in Figure 4, where on the left we have the certified guarantees for $N_{\mathrm{cert}} = 2000$ samples from ImageNet1K; on the right we have the empirical radii for $N_{\mathrm{emp}} = 250$ samples obtained by applying a standard box attack [33] strategy with $q = 16$. We observe from the certified results that the decremental stability radii are larger than those of incremental stability. This is reasonable since the base classifier sees much more of the input when analyzing decremental stability and is thus more confident on average, i.e., achieves a larger confidence gap. Moreover, our empirical radii often cover up to half of the input, which suggests that our certified analysis is quite conservative.

## 4.2 (E2) Stability-Accuracy Trade-Offs

We next investigate how smoothing impacts the classifier accuracy. As $\lambda$ decreases due to more smoothing, the base classifier sees increasingly zeroed-out features — which should hurt accuracy. We took $N = 2000$ samples for each classifier on their respective datasets and plotted the certified accuracy vs. radius of decremental stability.

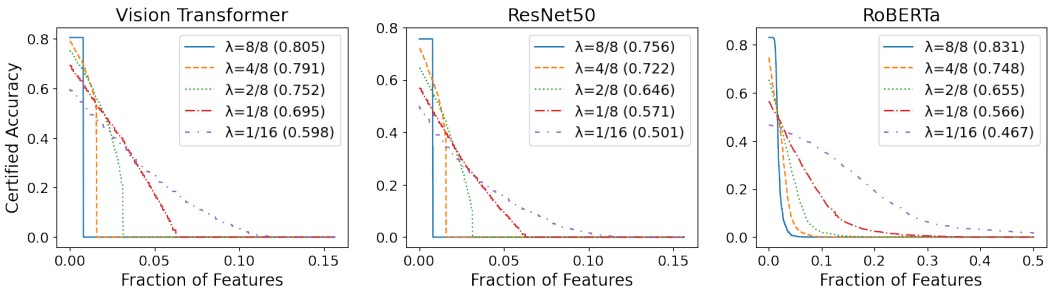

Figure 5: Certified accuracy vs. decremental stability radius. $N = 2000$.

We show the results in Figure 5, where the clean accuracy (in parentheses) decreases with $\lambda$ as expected. This accuracy drop is especially pronounced for ResNet50, and we suspect that the transformer architecture of Vision Transformer and RoBERTa makes them more resilient to the randomized masking of features. Nevertheless, this experiment demonstrates that large models, especially transformers, can tolerate non-trivial noise from MuS while maintaining high accuracy.

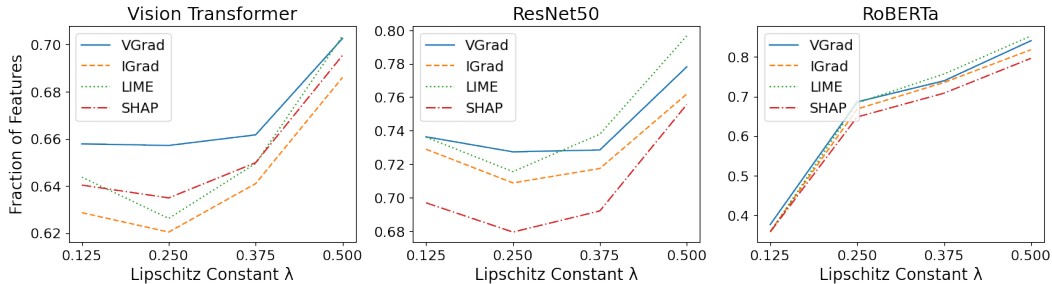

Figure 6: Average $k/n$ vs. $\lambda$, where $k = \|\varphi(x)\|_1$ is the number of features for $\langle f, \varphi \rangle$ to be consistent, incrementally stable with radius 1, and decrementally stable with radius 1. $N = 250$.

### 4.3 (E3) Which Explanation Method to Pick?

Finally, we explore which feature attribution method is best suited to stability guarantees of explainable model $\langle f, \varphi \rangle$. All four methods $\psi \in \{\text{LIME}, \text{SHAP}, \text{VGrad}, \text{IGrad}\}$ are continuous-valued, for which we samples $N = 250$ inputs from each model's respective dataset. For each input $x$, we use the feature importance ranking generated by $\psi(x)$ to iteratively build $\varphi(x)$ in a greedy manner like in Section 2.4. For some $x$, let $k_x = \varphi(x)/n$ be the number fraction of features needed for $\langle f, \varphi \rangle$ to be consistent, incrementally stable with radius 1, and decrementally stable with radius 1. We then plot the average $k_x$ for different methods at $\lambda \in \{1/8, \ldots, 4/8\}$ in Figure 6, where note that SHAP tends to require fewer features to achieve the desired properties, while VGrad tends to require more. However, we do not believe these to be decisive results, as many curves are relatively close, especially for Vision Transformer and ResNet50.

## 5  Related Works

For extensive surveys on explainability methods see [16, 20, 34, 35, 36, 37, 38]. Notable feature attribution methods include Vanilla Gradient Saliency [5], SmoothGrad [23], Integrated Gradients [8], Grad-CAM [39], Occlusion [40], LIME [6], SHAP [7], and their variants. Of these, Shapley value-based [17] methods [7, 24, 25] are rooted in axiomatic principles, as are Integrated Gradients [8, 41]. The work of [42] finds confidence intervals over attribution scores. A study of common feature attribution methods is done in [43]. Similar to our approach is [44], which studies binary-valued classifiers and presents an algorithm with succinctness and probabilistic precision guarantees. Different metrics for evaluating feature attributions are studied in [16, 18, 45, 46, 47, 48, 49, 50, 51]. Whether an attribution correctly identifies relevant features is a well-known issue [52, 53]. Many methods are also susceptible to adversarial attacks [54, 55]. As a negative result, [56] shows that

feature attributions have provably poor performance on sufficiently rich model classes. Related to feature attributions are *data attributions* [57, 58, 59], which assigns values to training data points. Also related to formal guarantees are formal methods-based approaches towards explainability [60].

## 6    Conclusion

We study provable stability guarantees for binary feature attribution methods through the framework of explainable models. A selection of features is stable if the additional inclusion of other features does not alter its explanatory power. We show that if the classifier is Lipschitz with respect to the masking of features, then one can guarantee relaxed variants of stability. To achieve this Lipschitz condition, we develop a smoothing method called Multiplicative Smoothing (MuS). This method is parametric to the choice of noise distribution, allowing us to construct and exploit distributions with structured dependence for exact and efficient evaluation. We evaluate MuS on vision and language models and demonstrate that MuS yields strong stability guarantees at only a small cost to accuracy.

## Acknowledgements

This research was partially supported by NSF award SLES 2331783 and a gift from AWS AI.

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

# A Proofs and Theoretical Discussions

In this Section, we present the proofs of our main results and some extensions to MuS.

## A.1 Proofs of Main Results

### A.1.1 Proof of Theorem 3.2

*Proof.* By linearity, we have:

$$g(x, \alpha) - g(x, \alpha') = \mathop{\mathbb{E}}_{s \sim \mathcal{D}} h(x \odot \tilde{\alpha}) - h(x \odot \tilde{\alpha}'), \qquad \tilde{\alpha} = \alpha \odot s, \qquad \tilde{\alpha}' = \alpha' \odot s,$$

so it suffices to analyze an arbitrary term by fixing some $s \sim \mathcal{D}$. Consider any $x \in \mathcal{X}$, let $\alpha, \alpha' \in \{0, 1\}^n$, and define $\delta = \alpha - \alpha'$. Observe that $\tilde{\alpha}_i \neq \tilde{\alpha}'_i$ exactly when $|\delta_i| = 1$ and $s_i = 1$. Since $s_i \sim \mathcal{B}(\lambda)$, we thus have $\Pr[\tilde{\alpha}_i \neq \tilde{\alpha}'_i] = \lambda |\delta_i|$, and applying the union bound:

$$\Pr_{s \sim \mathcal{D}} [\tilde{\alpha} \neq \tilde{\alpha}'] = \Pr_{s \sim \mathcal{D}} \left[ \bigcup_{i=1}^n \tilde{\alpha}_i \neq \tilde{\alpha}'_i \right] \leq \sum_{i=1}^n \lambda |\delta_i| = \lambda \|\delta\|_1,$$

and so:

$$
\begin{aligned}
|g(x, \alpha) - g(x, \alpha')| &= \left| \mathop{\mathbb{E}}_{s \sim \mathcal{D}} [h(x \odot \tilde{\alpha}) - h(x \odot \tilde{\alpha}')] \right| \\
&= \left| \Pr_{s \sim \mathcal{D}} [\tilde{\alpha} \neq \tilde{\alpha}'] \cdot \mathop{\mathbb{E}}_{s \sim \mathcal{D}} [h(x \odot \tilde{\alpha}) - h(x \odot \tilde{\alpha}') \,|\, \tilde{\alpha} \neq \tilde{\alpha}'] \right. \\
&\quad \left. - \Pr_{s \sim \mathcal{D}} [\tilde{\alpha} = \tilde{\alpha}'] \cdot \mathop{\mathbb{E}}_{s \sim \mathcal{D}} [h(x \odot \tilde{\alpha}) - h(x \odot \tilde{\alpha}') \,|\, \tilde{\alpha} = \tilde{\alpha}'] \right|.
\end{aligned}
$$

Note that $\mathbb{E}[h(x \odot \tilde{\alpha}) - h(x \odot \tilde{\alpha}') \,|\, \tilde{\alpha} = \tilde{\alpha}'] = 0$, and so

$$
\begin{aligned}
|g(x, \alpha) - g(x, \alpha')| &= \Pr_{s \sim \mathcal{D}} [\tilde{\alpha} \neq \tilde{\alpha}'] \cdot \underbrace{\left| \mathop{\mathbb{E}}_{s \sim \mathcal{D}} [h(x \odot \tilde{\alpha}) - h(x \odot \tilde{\alpha}') \,|\, \tilde{\alpha} \neq \tilde{\alpha}'] \right|}_{\leq 1 \;\; \text{because } h(\cdot) \in [0, 1]} \\
&\leq \Pr_{s \sim \mathcal{D}} [\tilde{\alpha} \neq \tilde{\alpha}'] \\
&\leq \lambda \|\delta\|_1.
\end{aligned}
$$

Thus, $g(x, \cdot)$ is $\lambda$-Lipschitz in the $\ell^1$ norm. $\qquad \square$

### A.1.2 Proof of Theorem 3.3

*Proof.* We first show incremental stability. Consider any $x \in \mathcal{X}$, then by masking equivalence:

$$f(x \odot \varphi(x)) = g(x \odot \varphi(x), \mathbf{1}) = g(x, \varphi(x)),$$

and let $g_A, g_B$ be the top-two class probabilities of $g$ as defined in (1). By Theorem 3.2, both $g_A, g_B$ are Lipschitz in their second parameter, and so for all $\alpha \in \{0, 1\}^n$:

$$
\begin{aligned}
\|g_A(x, \varphi(x)) - g_A(x, \alpha)\|_1 &\leq \lambda \|\varphi(x) - \alpha\|_1 \\
\|g_B(x, \varphi(x)) - g_B(x, \alpha)\|_1 &\leq \lambda \|\varphi(x) - \alpha\|_1
\end{aligned}
$$

Observe that if $\alpha$ is sufficiently close to $\varphi(x)$, i.e.:

$$2\lambda \|\varphi(x) - \alpha\|_1 \leq g_A(x, \varphi(x)) - g_B(x, \varphi(x)),$$

then the top-class index of $g(x, \varphi(x))$ and $g(x, \alpha)$ are the same. This means that $g(x, \varphi(x)) \cong g(x, \alpha)$ and thus $f(x \odot \varphi(x)) \cong f(x \odot \alpha)$, thus proving incremental stability with radius $d(x, \varphi(x))/(2\lambda)$.

The decremental stability case is similar, except we replace $\varphi(x)$ with $\mathbf{1}$. $\qquad \square$

## A.2 Some Basic Extensions

Below, we present some extensions to MuS that help increase the fraction of the input to which we can guarantee stability.

### A.2.1 Feature Grouping

We have so far assumed that $\mathcal{X} = \mathbb{R}^n$, but sometimes it may be desirable to group features together, e.g., color channels of the same pixel. Our results also hold for more general $\mathcal{X} = \mathbb{R}^{d_1} \times \cdots \times \mathbb{R}^{d_n}$, where for such $x \in \mathcal{X}$ and $\alpha \in \mathbb{R}^n$ we lift $\odot$ as

$$\odot : \mathcal{X} \times \mathbb{R}^n \to \mathcal{X}, \qquad (x \odot \alpha)_i = x_i \cdot \mathbb{I}[\alpha_i = 1] \in \mathbb{R}^{d_i}.$$

All of our proofs are identical under this construction, with the exception of the dimensionalities of terms like $(x \odot \alpha)$. Figure 1 gives an example of feature grouping.

## B  All Experiments

**Models, Datasets, and Explanation Methods**  We evaluate on two vision models (Vision Transformer [9] and ResNet50 [29]) and one language model (RoBERTa [30]). For the vision dataset, we use ImageNet1K [31]; for the language dataset, we use TweetEval [32] sentiment analysis. We use four explanation methods in SHAP [7], LIME [6], Integrated Gradients (IGrad) [8], and Vanilla Gradient Saliency (VGrad) [5]; where we take $\varphi(x)$ as the top-$k$ weighted features.

**Training Details**  We used Adam [61] as our optimizer with default parameters and a learning rate of $10^{-6}$ for 5 epochs. Because we consider $\lambda \in \{1/8, 2/8, 3/8, 4/8, 8/8\}$ and $h \in \{\text{Vision Transformer}, \text{ResNet50}, \text{RoBERTa}\}$, there are a total of 15 different models for most experiments. To train with a particular $\lambda$: for each training input $x$, we generate two random maskings — one where $\lambda$ of the features are zeroed and one where $\lambda/2$ of the features are zeroed. This additional $\lambda/2$ zeroing is to account for the fact that inputs to a smoothed model will be subject to masking by $\lambda$ as well as $\varphi(x)$, where the scaling factor of $1/2$ is informed by our prior experience about the size of a stable explanation.

**Miscellaneous Preprocessing**  For images in ImageNet1K we use feature grouping (Section A.2.1) to group the $3 \times 224 \times 224$ dimensional image into patches of size $3 \times 28 \times 28$, such that there remains $n = 64$ feature groups. Each feature of a feature group then receives the same value of noise during smoothing. We report radii of stability as a fraction of the feature groups covered. For example, if at some input from ImageNet1K, we get an incremental stability radius of $r$, then we report $r/64$ as the fraction of features up to which we are guaranteed to be stable. This is especially amenable to evaluating RoBERTa on TweetEval where inputs do not have uniform token lengths, i.e., do not have uniform feature dimensions. In all of our experiments, we use the quantized noise as in Section 3.3 with a quantization parameter of $q = 64$, with the exception of Appendix B.2 and Appendix B.3. Our experiments are organized as follows:

- (Section B.1) What is the quality of stability guarantees?
- (Section B.2) What is the theoretical vs empirical stability that can be guaranteed?
    - We $q = 64$ for theoretical guarantees, $q = 16$ for empirical guarantees.
- (Section B.3) What are the stability-accuracy trade-offs?
    - We use $q \in \{16, 32, 64, 128\}$ to study the effect of $q$ on MuS-smoothed performance.
- (Section B.4) Which explanation method is best?
- (Section B.5) Does additive smoothing improve empirical stability?
- (Section B.6) Does adversarial training improve empirical stability?

### B.1  Quality of Stability Guarantees

Here we study what radii of stability are certifiable, and how often these can be achieved with different models and explanation methods. We therefore consider explainable models $\langle f, \varphi \rangle$ constructed from base models $h \in \{\text{Vision Transformer}, \text{ResNet50}, \text{RoBERTa}\}$ and explanation methods $\varphi \in \{\text{SHAP}, \text{LIME}, \text{IGrad}, \text{VGrad}\}$ with top-$k \in \{1/8, 2/8, 3/8\}$ feature selection. We take $N = 2000$ samples from each model's respective datasets and compute the following value for each radius:

$$\text{value}(r) = \frac{\#\{x : \langle f, \varphi \rangle \text{ consistent and inc (dec) stable with radius} \le r\}}{N}.$$

Plots of incremental stability are on the left; plots of decremental stability are on the right.

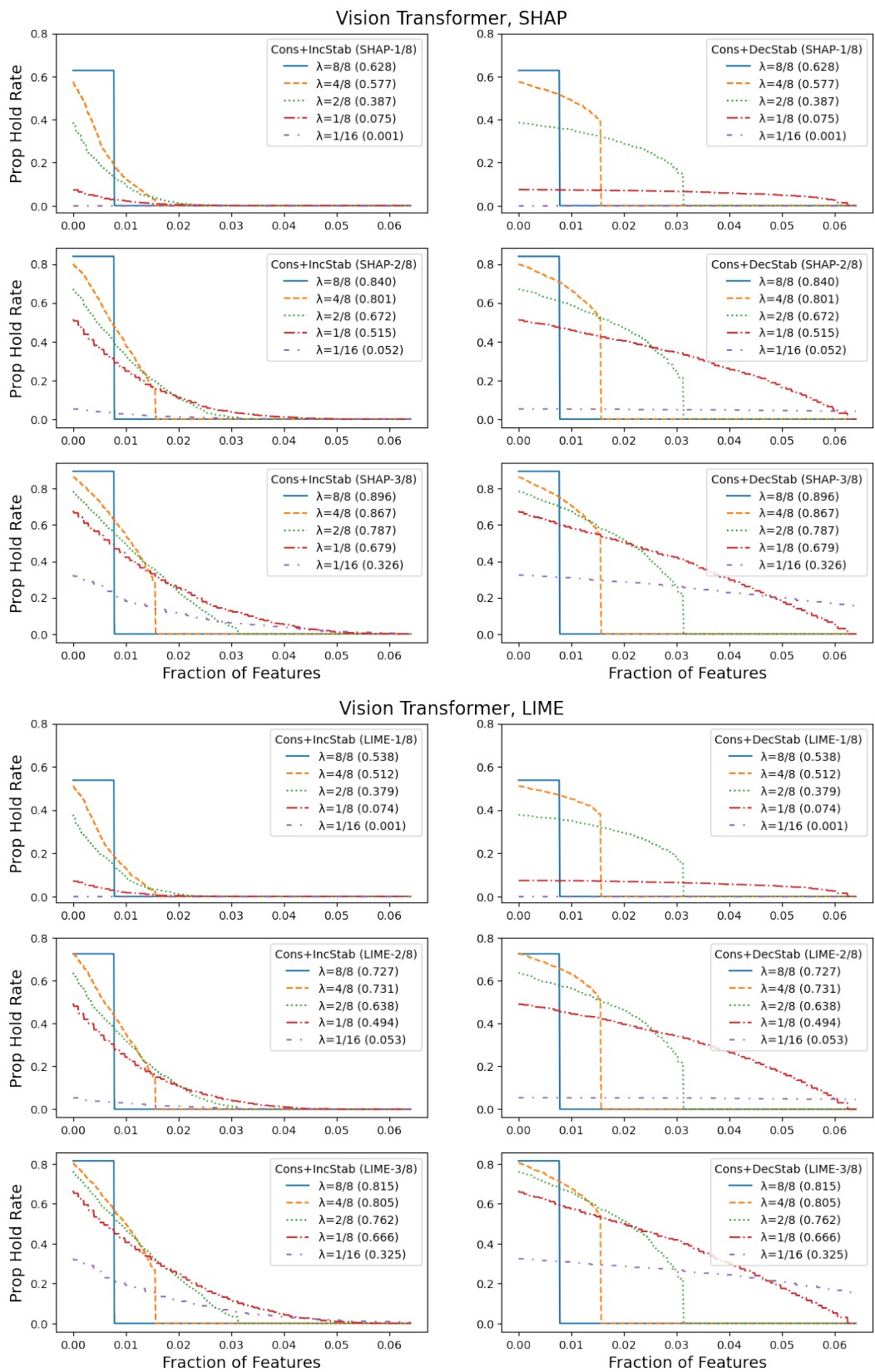

Figure 7: (Top) Vision Transformer with SHAP. (Bottom) Vision Transformer with LIME. (Left) consistent and incrementally stable. (Right) consistent and decrementally stable.

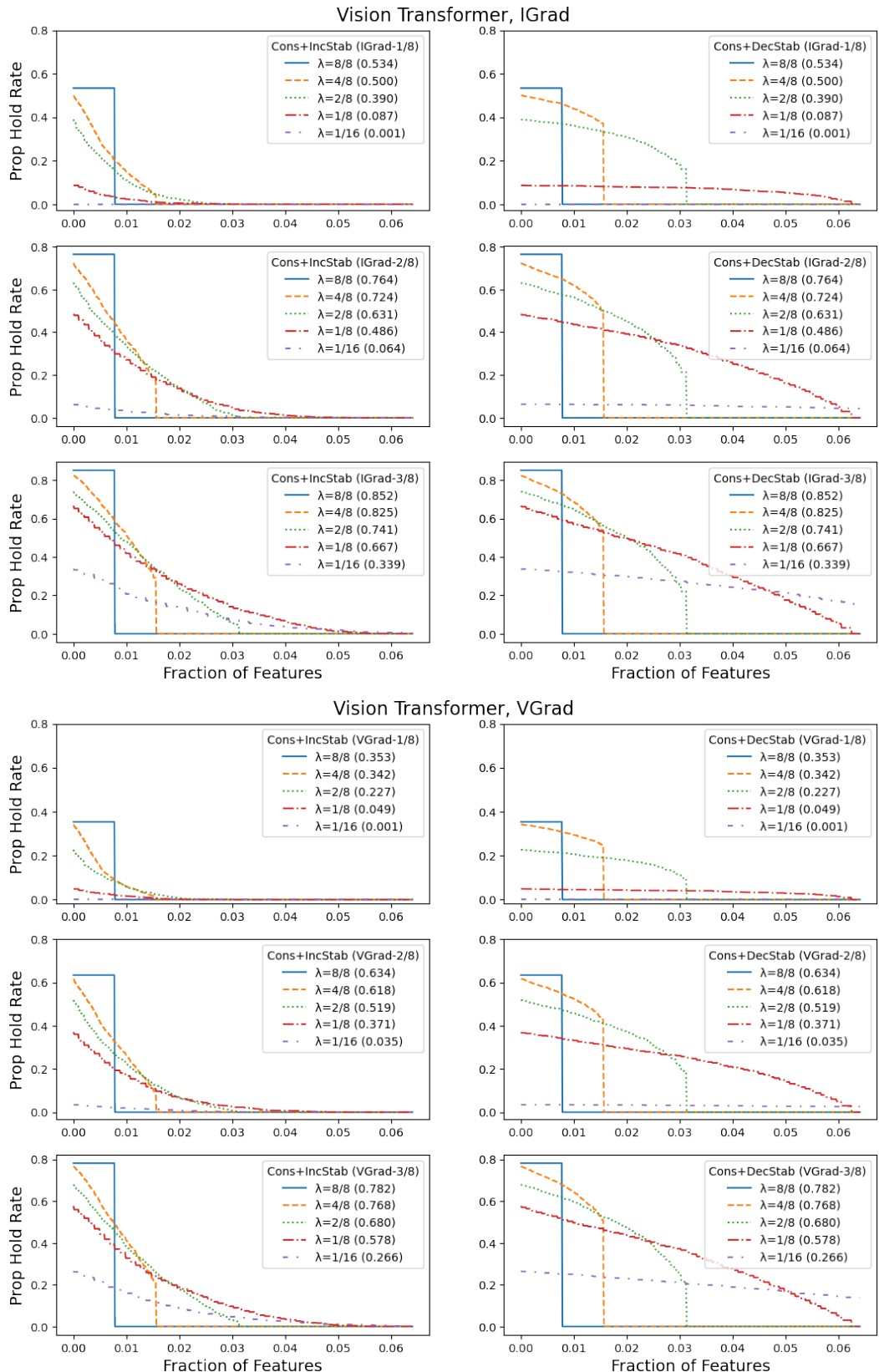

Figure 8: (Top) Vision Transformer with IGrad. (Bottom) Vision Transformer with VGrad. (Left) consistent and incrementally stable. (Right) consistent and decrementally stable.

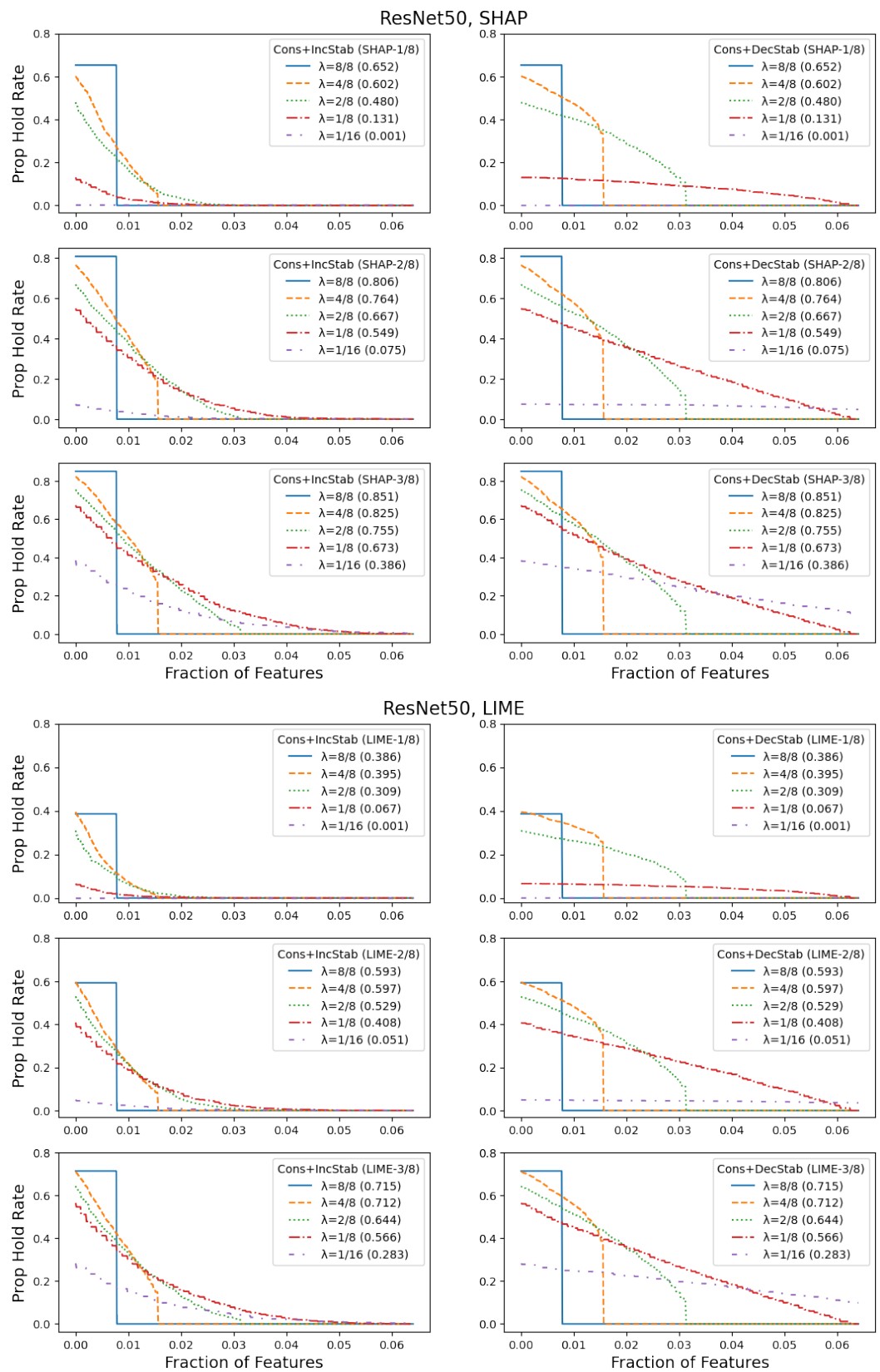

Figure 9: (Top) ResNet50 with SHAP. (Bottom) ResNet50 with LIME. (Left) consistent and incrementally stable. (Right) consistent and decrementally stable.

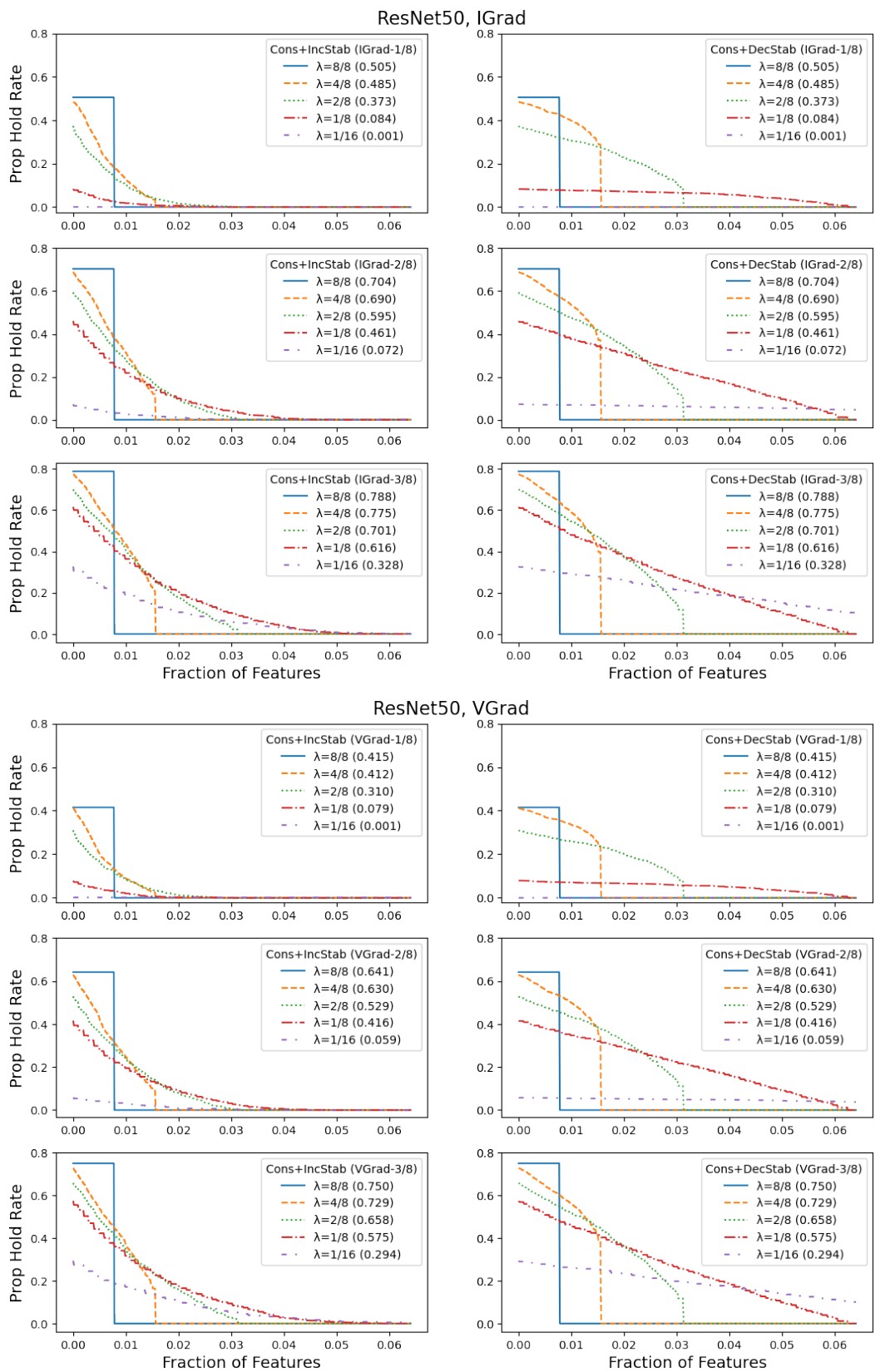

Figure 10: (Top) ResNet50 with IGrad. (Bottom) ResNet50 with VGrad. (Left) consistent and incrementally stable. (Right) consistent and decrementally stable.

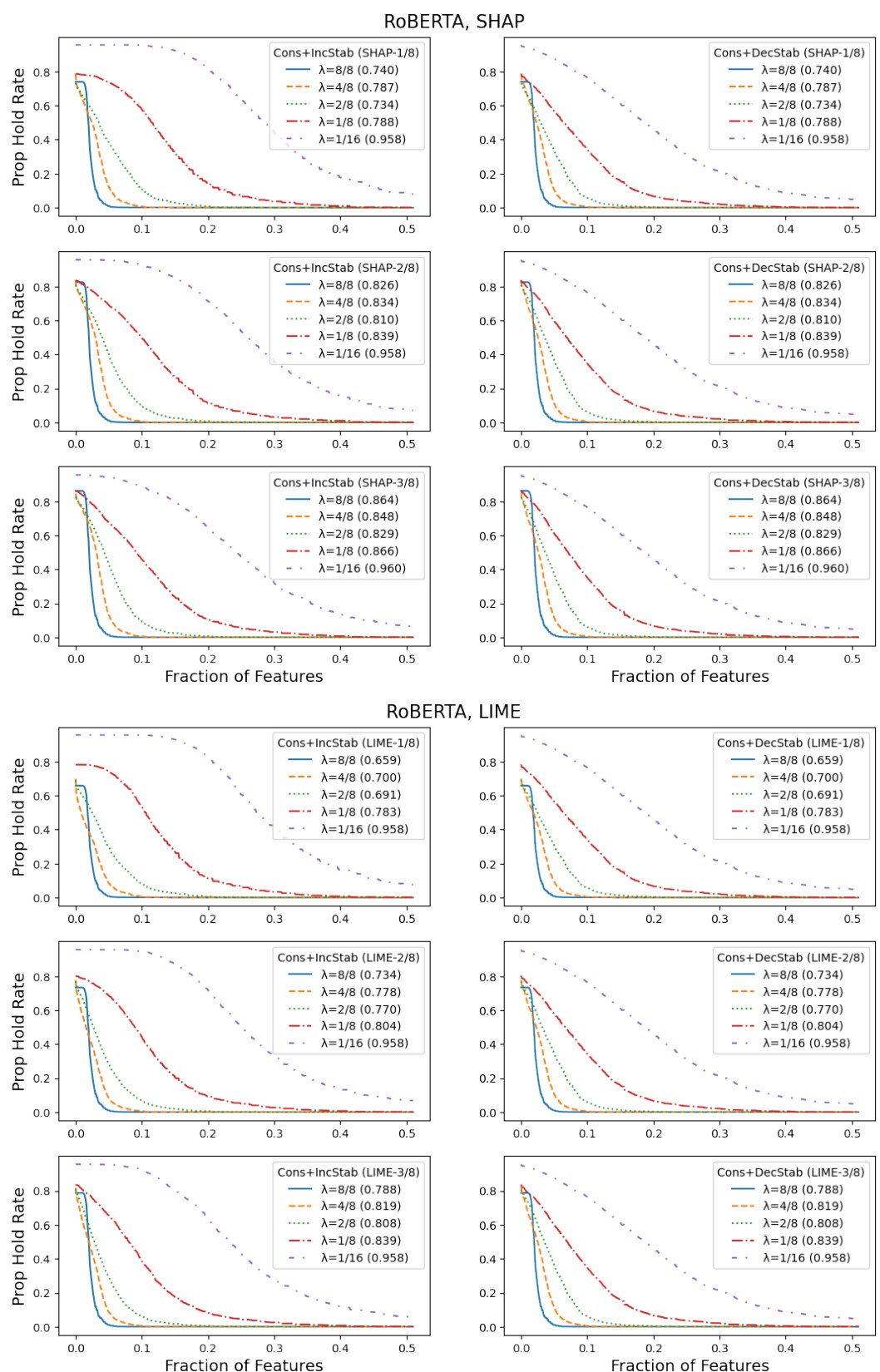

Figure 11: (Top) RoBERTa with SHAP. (Bottom) RoBERTa with LIME. (Left) consistent and incrementally stable. (Right) consistent and decrementally stable.

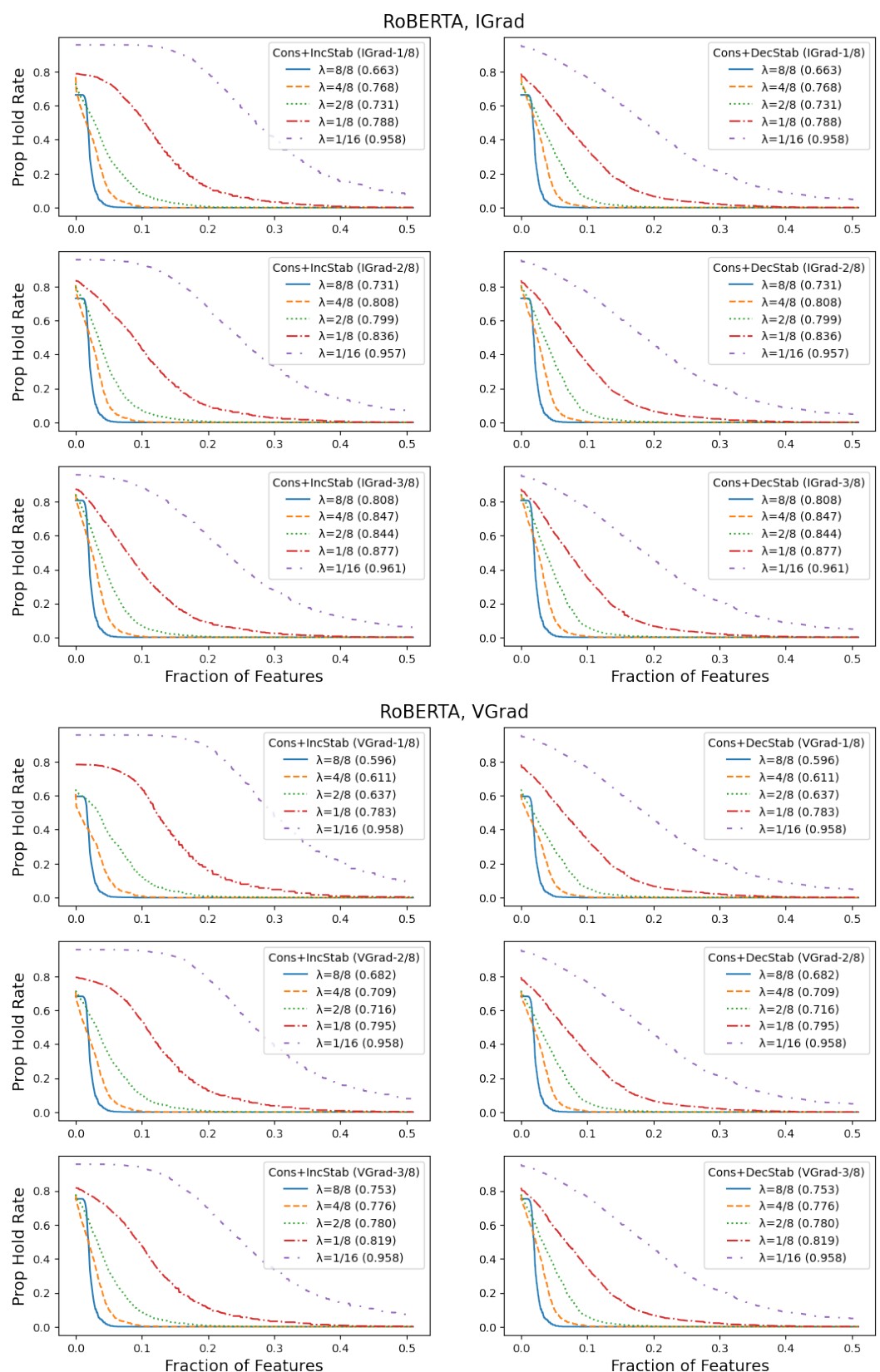

Figure 12: (Top) RoBERTa with IGrad. (Bottom) RoBERTa with VGrad. (Left) consistent and incrementally stable. (Right) consistent and decrementally stable.

## B.2 Theoretical vs Empirical

We compare the certifiable theoretical stability guarantees with what is empirically attained via a standard box attack search [33]. This is an extension of Section 4.2, where we now show all models as evaluated with SHAP-top25%. We take $q = 64$ with $N_{cert} = 2000$ for the certified plots, and $q = 64$ with $N_{emp} = 250$ for the empirical plots. This is because box attack is time-intensive.

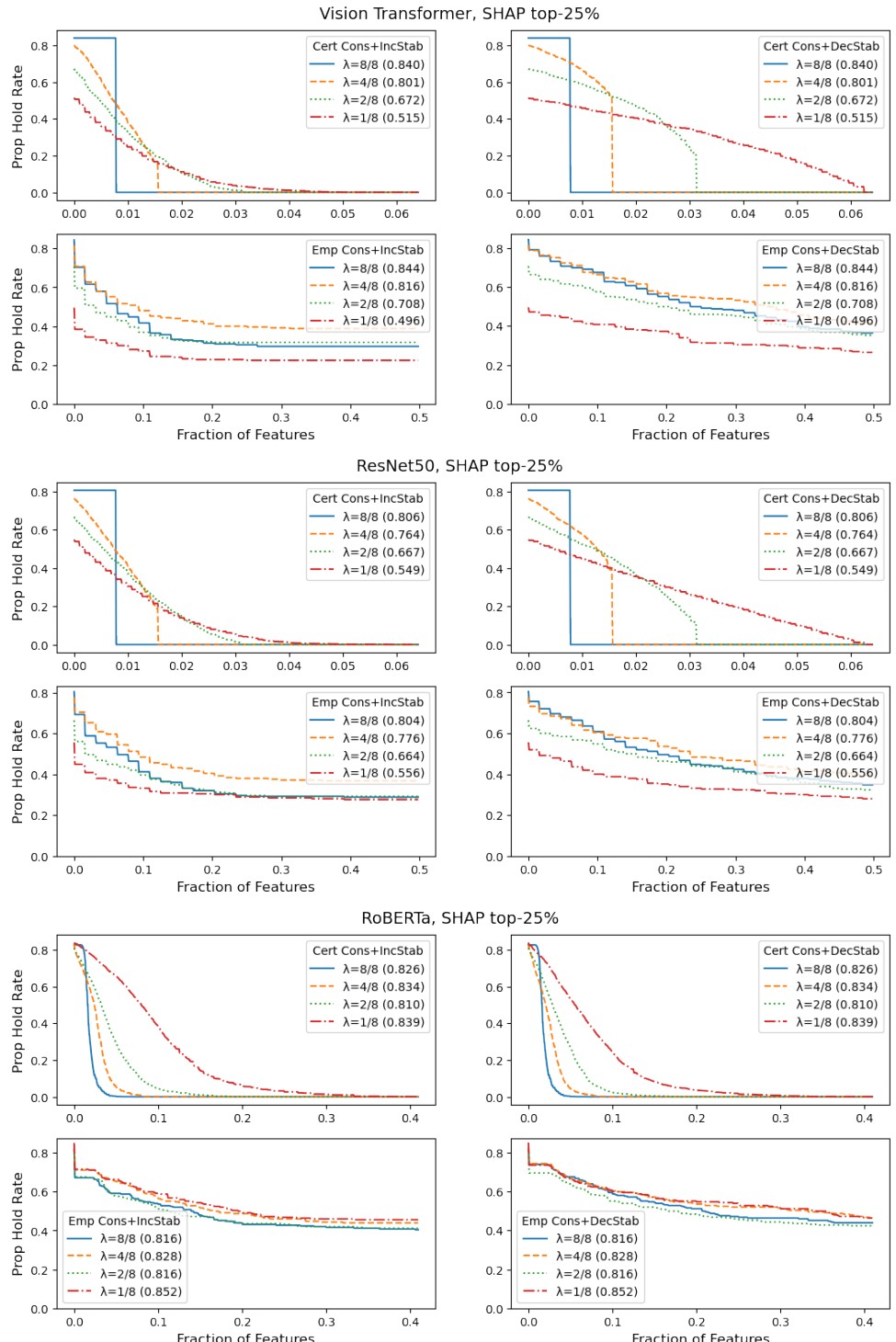

Figure 13: With SHAP top-25%. (Top) Vision Transformer. (Middle) ResNet50. (Bottom) RoBERTa.

## B.3 Stability-Accuracy Trade-Offs

We study how the accuracy degrades with $\lambda$. We consider a smoothed model $f$ constructed from a base classifier $h \in \{\text{Vision Transformer}, \text{ResNet50}, \text{RoBERTa}\}$ and vary $\lambda \in \{1/16, 1/8, 2/8, 4/8, 8/8\}$. We then take $N = 2000$ samples from each respective dataset and measure the accuracy of $f$ at different radii. We use $f(x) \cong \texttt{true\_label}$ to mean that $f$ attained the correct prediction at $x \in \mathcal{X}$, and we plot the following value at each radius $r$:

$$\text{value}(r) = \frac{\#\{x : f(x) \cong \texttt{true\_label} \text{ and dec stable with radius} \leq r\}}{N}$$

Below, we show the plots for different quantization parameters of $q \in \{16, 32, 64, 128\}$. The $q = 64$ plots are identical to that of Figure 5. We see that increasing $q$ generally improves the performance of MuS, but at a computational cost since this requires $q$ evaluations of $h$.

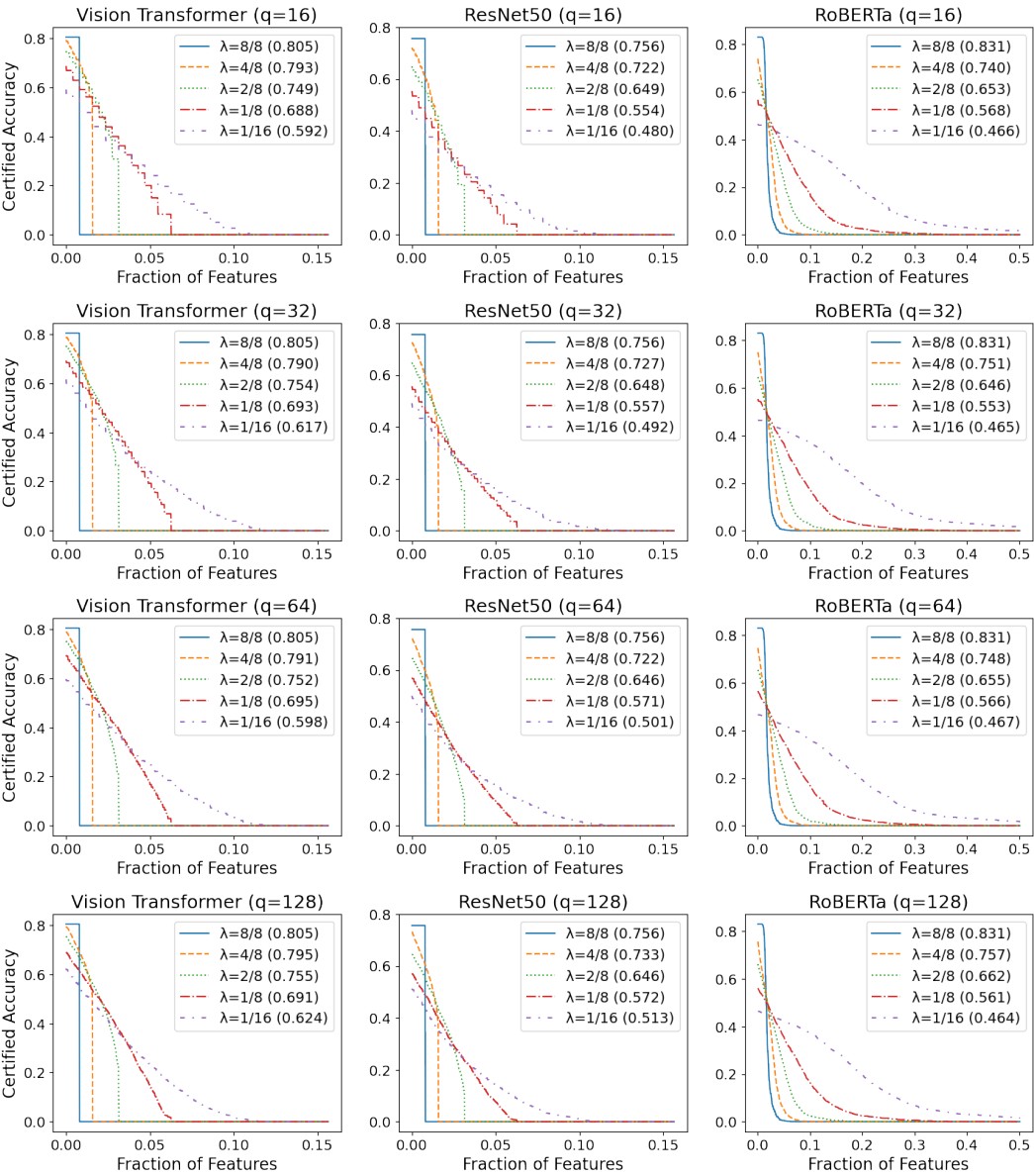

Figure 14: Certified accuracy plots for different quantization parameters of $q \in \{16, 32, 64, 128\}$.

## B.4 Which Explanation Method is the Best?

We first investigate how many features are needed to yield consistent and non-trivially stable explanations, as done by the greedy selection algorithm in Section 2.4. For some $x \in \mathcal{X}$, let $k_x$ denote the fraction of features that $\langle f, \varphi \rangle$ needs to be consistent, incrementally stable, and decrementally stable with radius 1. We vary $\lambda \in \{1/8, \ldots, 4/8\}$, where recall $\lambda \leq 4/8$ is needed for non-trivial stability, and use $N = 250$ samples to plot the average $k_x$. This part is identical to Section 4.3.

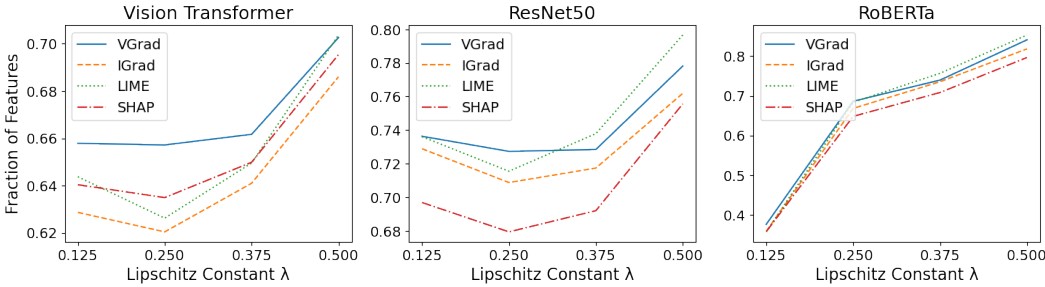

Figure 15: (Left) Vision Transformer. (Middle) ResNet50. (Right) RoBERTa.

We next investigate the ability of each method to predict features that lead to high accuracy. Let $f(x \odot \varphi(x)) \cong \texttt{true\_label}$, mean that the masked input $x \odot \varphi(x)$ yields the correct prediction. We then plot this accuracy as we vary the top-$k \in \{1/8, 2/8, 3/8\}$ for different methods $\varphi$, and $\lambda \in \{1/8, \ldots, 8/8\}$, using $N = 2000$ samples.

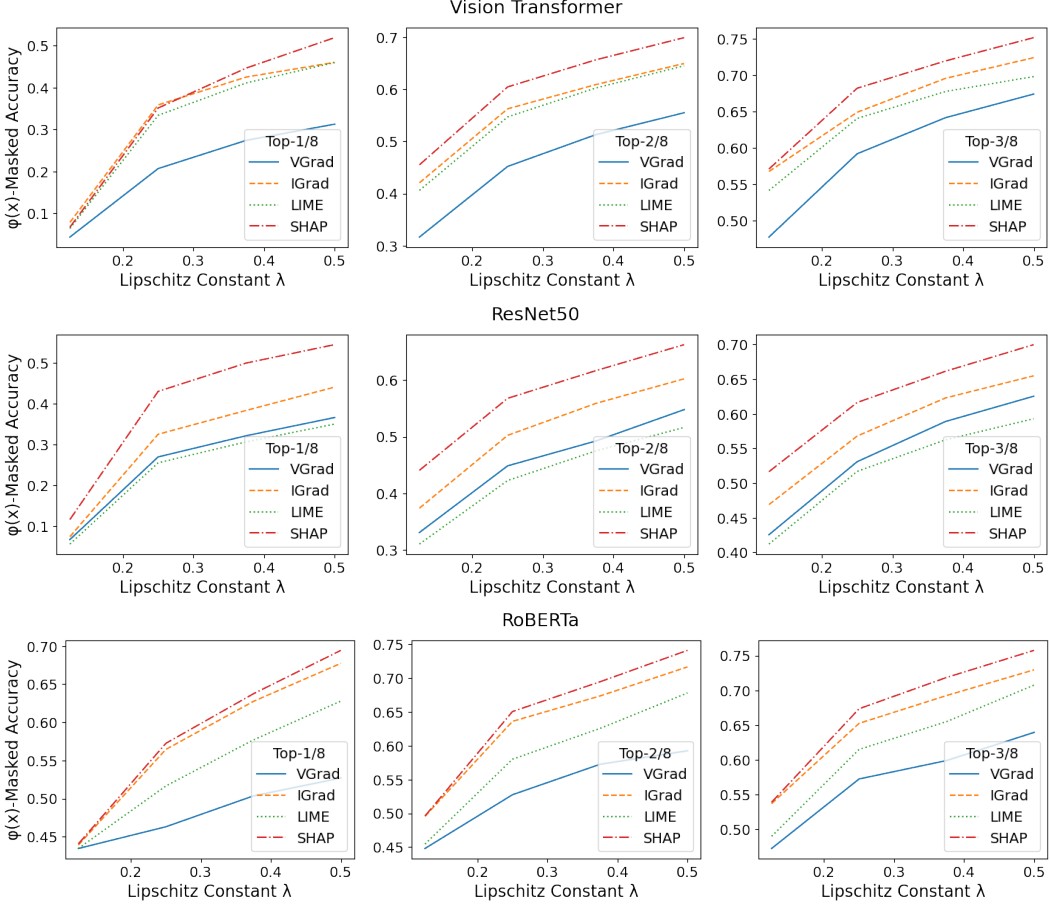

Figure 16: (Top) Vision Transformer. (Middle) ResNet50. (Bottom) RoBERTa.

## B.5 Empirical Stability of Additive Smoothing

We now study how well additive smoothing impacts empirical robustness compared to multiplicative smoothing. Although additive smoothing cannot yield theoretical stability guarantees (c.f. Proposition 3.1), we nevertheless investigate its empirical performance. We use explanations generated by SHAP top-25% and use Vision Transformer as our base model $h$. We fine-tuned different variants of Vision Transformer at $\lambda \in \{8/8, 4/8, 2/8, 1/8\}$ and used these in two general classes of models: Vision Transformer with multiplicative smoothing, and Vision Transformer with additive smoothing. We call these two $f_{\text{mus}}$ and $f_{\text{add}}$ respectively, and define them as follows:

$$f_{\text{mus}}(x) = \mathop{\mathbb{E}}_{s \sim \mathcal{D}} h(x \odot s), \qquad f_{\text{add}}(x) = \mathop{\mathbb{E}}_{s \sim \mathcal{U}(-1/2\lambda, 1/2\lambda)} h(x + s)$$

where for multiplicative smoothing, $\mathcal{D}$ is as in Theorem 3.2. Over $N = 250$ samples from ImageNet1K, we check how often incremental stability of radius $\geq 1$ is obtained, and plot our results in Figure 17 (left). We see that multiplicative smoothing yields better empirical performance than additive smoothing.

## B.6 Empirical Stability of Adversarial Training

Similar to Section B.5, we also check how adversarial training [62] affects empirical stability. We consider ResNet50 with different adversarial training setups from the Robustness Python library [63] and compare them to their respective MuS-wrapped variants. As with Section B.5 we take $N = 250$ samples from ImageNet1K, and plot our results in Figure 17 (right).

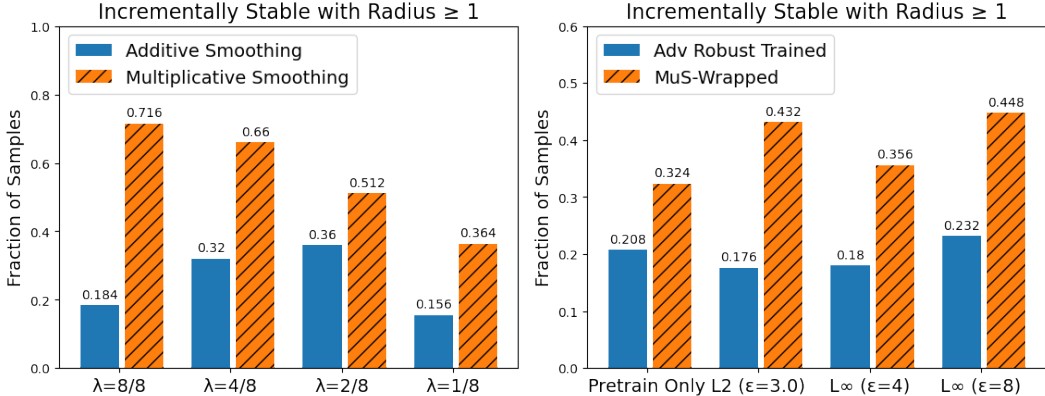

Figure 17: How often does each model empirically attain incremental stability of radius $\geq 1$? We check every $\alpha' \succeq \alpha$ where $\|\alpha' - \alpha\|_1 = 1$. (Left) Additive smoothing vs. multiplicative smoothing. (Right) ResNet50 with different adversarial training setups vs. their respective MuS-wrapped variants.

**B.7  Discussion**

**Effect of Smoothing**  We observe that smoothing can yield non-trivial stability guarantees, especially for Vision Transformer and RoBERTa, as evidenced in Appendix B.1. We see that smoothing is least detrimental on these two transformer-based architectures, and most negatively impacts the performance of ResNet50. We conjecture that although different training set-ups may improve performance across every category, this still serves to illustrate the general trend.

**Theoretical vs Empirical**  It is expected that the certifiable radii of stability is more conservative than what is empirically observed. As mentioned in Section 3.2, for each $\lambda$, there is a maximum radius to which stability can be guaranteed, which is an inherent limitation of using confidence gaps and Lipschitz constants as the main theoretical technique. We emphasize that the notion of stability need not be tied to smoothing, though we are currently not aware of other viable approaches.

**Why these Explanation Methods?**  We chose SHAP, LIME, IGrad, and VGrad from among the large variety of methods available primarily due to their popularity, and because we believe that they are collectively representative of many techniques. In particular, we believe that LIME remains a representative baseline for surrogate model-based explanation methods. SHAP and IGrad are, to our knowledge, the two most well-known families of axiomatic feature attribution methods. Finally, we believe that VGrad is representative of a traditional gradient saliency-based approach.

**Which Explanation Method is the Best?**  Based on our experiments in Appendix B.4 we see that SHAP generally achieves higher accuracy using the same amount of top-$k$ features as other methods. On the other hand, VGrad tends to perform poorly. We remark that there are well-known critiques against the usefulness of saliency-based explanation methods [53].

# C  Miscellaneous

**Relevance to Other Explanation Methods**  Our key theoretical contribution of MuS in Theorem 3.2 is a general-purpose smoothing method that is distinct from standard smoothing techniques, namely additive smoothing. Therefore, MuS is applicable to other problem domains beyond what is studied in this paper and would be useful where small Lipschitz constants with respect to maskings are desirable.

**Broader Impacts**  Reliable explanations are necessary for making well-informed decisions and are increasingly important as machine learning models are integrated with fields like medicine, law, and business — where the primary users may not be well-versed in the technical limitations of different methods. Formal guarantees are thus important for ensuring the predictability and reliability of complex systems, which then allows users to construct accurate mental models of interaction and behavior. In this work, we study a particular kind of guarantee known as stability, which is key to feature attribution-based explanation methods.

