# OpenReview forum: "Stability Guarantees for Feature Attributions with Multiplicative Smoothing"
_NeurIPS.cc/2023/Conference — NeurIPS 2023 poster_

### Official Review · Reviewer_WB7f · 2023-07-01

**Soundness:** 3 good
**Presentation:** 2 fair
**Contribution:** 2 fair
**Rating:** 6
**Confidence:** 3

**Summary:**

The paper is about the stability of explanations in the feature attribution setting for image classification tasks. They illustrate that in certain settings, swapping/removing a pixel of an explanation completely changes the classifiers prediction, which is undesired.

They define what a "stable" explanation is formally and introduce Multiplicative Smoothing as a method to obtain these stable explanations. In principle it's multiplying the image with feature attribution masks such that the classifier becomes smooth in the sense that it doesn't suddenly (non-smoothly) change it's prediction from for example cat to goldfish.

Experiments demonstrate the efficacy of MuS.

**Strengths:**

1. It's clear what the paper is trying to achieve
2. Solution proposal is simple and thoroughly explained with good theoretical justification.


**Weaknesses:**

1. It's a bit unclear what exactly smoothness refers to, currently it seems to refer to no discontinuous jumps when removing a feature attribution pixel in classification output?
2. The experiments are a bit unintuitive, needs clarification what's being tested here and why.



**Questions:**

1. What's smoothness exactly, never got a good intuition what this exactly means.
2. Don't understand what the experiments do, please explain how and why these experiments demonstrate that MuS is working as intended.
3. Is it correct to understand that the paper is in some sense looking for the "minimal" set up features that explains a class that won't change with some perturbation ?

-----

Post rebuttal: Authors did a good job explaining what's going on, left some feedback for exposition, raised to 6, it's an interesting paper.

---

> ### Author Rebuttal · Authors · 2023-08-10
>
> We thank the Reviewer for their time and comments. We will accordingly update our exposition to better clarify our methods, experiments, and contributions. Below we group some of the Reviewer's comments and questions in our response.
>
> ### Weakness 1 / Question 1: Meaning of smoothness.
>
> Each coordinate of our smoothed classifier is $\lambda$-Lipschitz with respect to the binary attribution $\alpha$ in the $L^1$ norm. Recall that on binary vectors the $L^1$ metric measures bit flips. We will add an explicit definition of this in Section 2.3, and clarify this in the discussion of Theorem 3.2.
>
> Lipschitz smoothness means that whenever we mask/unmask a few features of the input $x$, the change in classifier confidence is provably bounded. In other words: if we fix $x$ and consider two binary attributions $\alpha$ and $\alpha'$, the prediction confidences of $f(x \odot \alpha)$ and $f(x \odot \alpha')$ will be similar provided that $\alpha$ and $\alpha'$ do not differ on too many bits.
>
> If the predicted class of $f(x \odot \alpha)$ is substantially more confident than the second-best class, then using any other sufficiently similar binary attribution $\alpha'$ will induce the same class. That is, $f(x \odot \alpha)$ and $f(x \odot \alpha')$ will provably yield the same class if: (1) there is a sufficient confidence gap between the first-best and second-best class, and (2) if $\alpha$ and $\alpha'$ do not differ by too many bit flips. This condition is computationally fast to check if $f$ is Lipschitz smooth, and we use MuS to guarantee Lipschitz smoothness in a sample-efficient and deterministic manner.
>
> ### Weakness 2 / Question 2: Clarity of experiments.
>
> We thank the Reviewer for raising this concern. We will amend our experiments section to better clarify each experiment.
>
> **Experiment 1 (Section 4.1):** This experiment shows that off-the-shelf classifier-explainer pairs can indeed achieve non-trivial stability guarantees. This is a significant result since there did not previously exist any stability-like guarantees for explanations. Moreover, our chosen classifiers and explanation methods were not customized for stability, so it is remarkable that MuS can still prove non-trivial stability in a large number of cases without much custom engineering. As our proposed method is classifier and explainer-agnostic, we do not investigate new model architectures nor explanation methods in this paper, and instead opt to use existing ones to test MuS.
>
> **Experiment 2 (Section 4.2):** This experiment evaluates the accuracy degradation of the smoothed classifier as a function of $\lambda$. Because smoothing methods inject noise when evaluating the original ("base") classifier, it is expected that greater smoothness (i.e. smaller Lipschitz constants) is achieved at a cost to smoothed-classifier accuracy. Much of smoothing literature, ours included, is concerned with navigating this smoothness-accuracy trade-off. Figure 5 (see also Section 2 of Rebuttal Supplementals) shows the certified accuracy curve of different classifiers under different amounts of noise, where the far-left value (in parentheses) is the overall accuracy of the smoothed classifier. In particular, the $\lambda=1/16$ curve is effectively a stress test of when the smoothed classifier is subject to extreme amounts of noise, higher than what is common in smoothing literature. It is at these extreme noises that one begins to see non-trivial accuracy degradation in the smoothed model.
>
> **Experiment 3 (Section 4.3):** This experiment does not directly evaluate the efficacy of MuS. Rather, it evaluates which of our four chosen feature attribution methods (Vanilla Gradient Saliency, Integrated Gradients, LIME, SHAP) are best suited for achieving certified stability if naively applied. In particular, we interpret higher attribution values to mean more important features, and we iteratively select the next-most-important feature into $\alpha$ until a stopping condition is achieved (consistency, radius-1 incremental, and decremental stability). The resulting $\alpha$ is then a proxy for the relative "efficiency" or "density" (the $k/n$ measure) of our chosen attribution methods. As stability-like guarantees do not exist for either continuous or binary attribution methods, we opted to simply use the most popular ones (which happen to be continuous) as a proof of concept test for efficiency. We remark that the construction of this $\alpha$ is technically a feature selection method, but we emphasize that this is only intended for crudely testing our four chosen methods.
>
> ### Question 3: Objective of paper.
>
> We believe there is a misunderstanding, and we will revise our exposition to better clarify our story. The focus of our paper is on analyzing and proving guarantees given an explanation. Our goal is not to find a minimal feature selection, nor to present a feature selection algorithm. We are agnostic to the selection method, which we assume is given. We do this because there do not yet exist feature attribution methods with useful explanation-relevant formal guarantees, so it would be premature to overly fixate on any particular one.
>
> We study binary feature attributions and ask: what properties are desirable, which conditions allow us to provably satisfy these properties, and how can we achieve said conditions? Our desired properties are stability (Definition 2.2), incremental stability (Definition 2.3), and decremental stability (Definition 2.4). Because our notion of stability is hard to show in general, we focus on the latter two: the condition that allows us to provably guarantee incremental/decremental stability is Lipschitz smoothness (Remark 2.6 and Theorem 3.3). We achieve Lipschitz smoothness using MuS (Theorem 3.2). Along with a sample-efficient noise construction in Section 3.3, we present a computationally efficient method for proving when a binary attribution is incrementally/decrementally stable.

---

> > ### Comment · Reviewer_WB7f · 2023-08-16
> > **Re: Rebuttal**
> >
> > Thanks for the clarifications!
> >
> > It's a bit more clear now what you're trying to achieve, it would be appreciated if you could associate the definitions with perhaps more figures or illustrations providing the intuition for stability.
> >
> > Happy to buy in and raise to 6.

---

### Official Review · Reviewer_WjBG · 2023-07-02

**Soundness:** 3 good
**Presentation:** 3 good
**Contribution:** 3 good
**Rating:** 6
**Confidence:** 4

**Summary:**

This paper aims to make the classifier robust to feature removal and addition. The authors find that adding a patch to the mask obtained from explanation method may cause the classifier to make substantially different prediction. They introduce the notion of incremental
stability and decremental stability to measure classifier's stability w.r.t mask change. They introduce MuS to any base classifier and show that MuS provides stability guarantee for this classifier.

**Strengths:**

1. The notion of incremental stability and decremental stability is novel.
2. They motivate multiplicative masking instead of the widely-adopted additive masking by pointing out the inconsistency involved in additive masking.
3. Certified stability guarantee is obtained.
4. A sample-efficient algorithm is introduced to reduce the sample complexity to sample from Bernoulli distribution.


**Weaknesses:**

1. Although masks can be sampled efficiently, to get the final prediction, a lot of inference steps are involved. This might be time-consuming. The authors do not talk about how this could be tackled.
2. MuS seems to be similar with Randomized Smoothing with Bernoulli distribution. By averaging predictions under several masks, the classifier use information from the whole image to make prediction. It is not too surprising that the classifier is more robust to mask change.

**Questions:**

1. Are robust models also sensitive to mask removal or addition?
2. There are many works discussing robust attributions (e.g., [1], [2]) and establishing the connection between the robustness of explanations and the robustness of black-box models (e.g., [3], [4]). What's the relationship between these works and this paper?

[1] Dombrowski et al., Explanations can be manipulated and geometry is to blame, NeurIPS 2019

[2] Wang et al., Smoothed Geometry for Robust Attribution, NeurIPS 2020

[3] Tan et al., Robust Explanation for Free or At the Cost of Faithfulness, ICML 2023

[4] Agarwal et al., Rethinking Stability for Attribution-based Explanations, ICLR 2022

**Limitations:**

Listed is Weakness and Questions sections.

---

> ### Author Rebuttal · Authors · 2023-08-10
>
> We thank the Reviewer for the helpful comments, questions, and references. We will include additional exposition and discussion to better clarify our sample complexity, especially relative to other smoothing methods. Moreover, we will include discussions on the listed references, which we believe will help us better clarify our position in the context of ML-explainability literature. Additionally, we have attached a Rebuttal Supplemental with material relevant to our response.
>
> ### Weakness 1: Sample efficiency of inference.
>
> The Reviewer is correct in identifying that requiring multiple model evaluations is a fundamental bottleneck for smoothing methods. We will emphasize in our revisions that our sample efficiency is significant relative to other smoothing methods: we require $<100$ samples to get deterministic guarantees whereas other smoothing methods often require $>10000$ samples to get probabilistic guarantees.
>
> ### Weakness 2: Similarity with Randomized Smoothing from Bernoulli Distribution.
>
> The Reviewer makes a good observation, and in fact, a Bernoulli distribution was one of our intermediate steps when developing MuS. We again emphasize that our formulation of MuS (Theorem 3.2) critically does not require coordinate-wise independence in its distribution (see also above discussion of Weakness 1). The advantage of this over a Bernoulli distribution is that we can then deterministically evaluate the smoothed classifier in $\ll 2^n$ samples via the construction in Section 3.3 (inspired from Reference [27]).
>
> We agree that for image classifiers especially, it is not surprising that a Bernoulli-like noising should preserve decent accuracy. Indeed our experiments, namely Figure 5, affirm this intuition (see also new version in Section 2 of Rebuttal Supplementals). Because using a Bernoulli distribution seemed so simple, we do not claim novelty on this idea alone, although we could not find other papers that explicitly discuss this. Instead, we only claim novelty on the distribution of Theorem 3.2 in conjunction with the sample-efficient construction in Section 3.3.
>
> ### Question 1: Robust model sensitivity to mask addition and removal.
>
> The Reviewer raises an interesting question, and we have run additional experiments with robust models. In particular, we consider four versions of pre-trained ResNet50 from the "robustness" Python package: non-robust ("base"), $L^2 (\varepsilon = 3.0)$, $L^\infty (\varepsilon = 4)$, and $L^\infty (\varepsilon = 8)$. As we cannot provably certify the stability of non-MuS models, we instead empirically test how often they are able to hit radius $r=1$ of incremental stability with and without using MuS. We use SHAP-top25% for the explanation method and take $N=250$ samples from ImageNet. Our $r=1$ incremental stability rates for the four models are listed below and also included in Section 3 of the Rebuttal Supplementals:
>
> Without MuS: 0.208, 0.176, 0.180, 0.232
>
> With MuS: 0.324, 0.432, 0.356, 0.448
>
> We observe that using MuS uniformly improves the empirical incremental stability rates.
>
> ### Question 2: Discussion with robust feature attributions.
>
> We thank the Reviewer for providing the references, and we will include discussions of these in our revised manuscript. An important difference is that we consider Lipschitz with respect to input maskings (expressed as $L^1$ Lipschitz in $\alpha$), whereas the listed references focus on adversarial robustness with respect to $L^p$ norms on continuous domains. This means that the domain and metric over which we define Lipschitz smoothness is fundamentally different than that of the listed references. An important consequence is that one unit of radius in our case easily achieves orders of magnitude more coverage on input features than that of the standard $L^p$-robustness case. For instance, in our examples and experiments we partition 3x224x224 images ($n=150528$ features) into 64 patches: one unit of radius in our setting then certifies 1/64 of the total 150k+ features. This is significantly more feature coverage compared to what is typical from $L^p$ smoothing-based methods, which often struggle to certify more than tens out of 150k+ features.
>
> We emphasize that this coverage improvement is due to our definition of Lipschitz-to-feature-maskings being fundamentally different from Lipschitz-to-$L^p$ norms on continuous spaces. Although $L^p$-robustness can in theory be used to certify properties of feature maskings, the certifiable radius is often too small to be useful. Crucially, our motivation to pursue this feature-masking variant of Lipschitz smoothness, and therefore MuS, is due to the insufficiency of common additive smoothing methods in our setting. This is because additive smoothing methods violate a condition that we call masking equivalence (Proposition 3.1), and therefore make them inadequate for use in the stability analysis of our context.

---

> > ### Comment · Reviewer_WjBG · 2023-08-13
> > **Response**
> >
> > Thanks for the detailed reply! I will raise my confidence score to 4. I have a few follow-up questions and I hope you can give some further explanations.
> >
> > 1. To obtain a prediction, how many inference steps are required? If ~100 inference steps were required, would the smoothing step be very slow?
> >
> > 2. In the experiment results you provide, it seems that with MuS, non-robust model has a higher hitting rate than $L_2 (\epsilon=3), L_\infty (\epsilon=4)$ trained robust models. Do you have any explanation on this result?

---

> > > ### Author Response · Authors · 2023-08-13
> > >
> > > **Q1. Cost of inference steps.**
> > >
> > > Yes. If $q=100$ samples are used, evaluating the smoothed model would be $\times 100$ slower than the base model. Our method for constructing the distribution allows us to decide the number of samples $q \geq 2$, and we arbitrarily set either $q = 64$ in most of our experiments and sometimes $q = 16$. In developmental testing, we have also found $q = 8$ to perform well (i.e. certified accuracy like Figure 5).
> > >
> > > Nevertheless, the Reviewer raises an interesting set of additional experiments to run for $q$ selection and we will include them in our revised manuscript. Below is a sample of the certified accuracy numbers (like Figure 5 / Section 2 of Rebuttal Supplementals, there both ran with $q=64$) when we run Vision Transformer at $\lambda = 1/8, 2/8, 3/8, 4/8$ for sample complexities of $q = 4,8, 16, 32, 64, 128$. On a $N = 2000$ sample from ImageNet we obtain the following smoothed classifier accuracies:
> > >
> > > $\lambda = 1/8$: N/A, 0.662, 0.688, 0.693, 0.695, 0.691
> > >
> > > $\lambda = 2/8$: 0.717, 0.743, 0.749, 0.754, 0.752, 0.755
> > >
> > > $\lambda = 3/8$: N/A, 0.760, 0.774, 0.777, 0.782, 0.778
> > >
> > > $\lambda = 4/8$: 0.776, 0.779, 0.793, 0.790, 0.791, 0.795
> > >
> > > Note that some values for $q = 4$ are N/A because this is an insufficient discretization granularity for specifying the corresponding probabilities at $\lambda = 1/8, 3/8$. We observe that in general, increasing the value of $q$ slightly increases the accuracy of the smoothed classifier. This may be because, heuristically, smaller values of $q$ mean that the smoothed classifier is more susceptible to the choice of initial random seed.
> > >
> > >
> > > **Q2. Robust vs non-robust model performance.**
> > >
> > > We were also surprised by the non-MuS performance of the different models, and we can only speculate as to the cause. One possibility is that some radii used in $L^p$ robustness training were too small to generalize to our setting. Our finding that $L^\infty (\varepsilon = 8)$ did in fact outperform the non-robust case is some preliminary evidence of this.

---

### Official Review · Reviewer_JJJj · 2023-07-11

**Soundness:** 3 good
**Presentation:** 3 good
**Contribution:** 2 fair
**Rating:** 6
**Confidence:** 2

**Summary:**

In this paper the authors have introduced a framework to measure the stability of feature attribution methods. They do so by introducing two relaxed notions of stability called incremental stability and descremental stability which check for stability in a neighbourhood of the original feature set. They show that size of these stable neighbourhoods can be measured for lipschitz smooth classification functions and then present a methdology to convert any classified into a Lipschitz smooth classifier using random sampling. Finally they conclude the paper with numerical experiments to illustrate the benefits of their approach.

**Strengths:**

I feel that this paper has several strengths.
Understanding the stability of feature attribution methods is a key requirement to evaluate them. This paper presents a novel and efficient approach to understanding these stability metrics which can then be used to compare quality of the features identified by the different methods. The stabiilty metric makes intuitive sense and the theoretical results justify the measurement techniques for them. They also present a way of how this stability metric can be extended to any classifier.
Overall, I believe this paper makes a key contribution in this area.


**Weaknesses:**

The metric seems more descriptive than prescriptive. It would be difficult to some how incorporate he metric into the optimization problem so as to guide the network and method towards more stable models and feature attributers.

**Questions:**

I think it is important to provide a discussion about what the impact of Mulitplicative smoothening is on the original classifier.
How is the stability metric of the smoothened classifier associated to the original classifier.


**Limitations:**

Adequately addressed.

---

> ### Author Rebuttal · Authors · 2023-08-10
>
> We thank the Reviewer for their time and comments. We will include additional discussion based on these comments, especially regarding how practitioners may apply and train for stability. Moreover, we remark that we have also attached a Rebuttal Supplemental document with additional examples and experiments. Below we address the Reviewer's comments.
>
> ### Weaknesses: Descriptive vs prescriptive metric; clarity for practitioners.
>
> The Reviewer makes a valid point that an optimization problem formulation is useful for practitioners looking to work with MuS and stability. We initially omitted an optimization formulation because we encountered a lot of difficulties in its specification, as any such formulation must consider many problem-dependent nuances, described below. Nevertheless, we believe that the Reviewer makes a strong case for the utility of an optimization formulation, even if in a restricted setting. We will include in the Appendix some formulations and also present discussions about implementation details. Below we further discuss the details and challenges of an optimization formulation for stability.
>
> The primary challenge with expressing stability metric as an optimization problem is that it is necessarily dependent on both the classifier model and explainer method. As such, any optimization formulation that maximizes stability should simultaneously train both the classifier and explainer. If one decides to parametrize and train the explainer, this opens a Pandora's box of feature attribution design, which is beyond the scope of our paper. If one fixes the explainer and optimizes only the model, then the optimization formulation is easier to write, although this is a more restrictive case as it neglects the explainer design. Nevertheless, we agree that the latter would be useful for practitioners looking to apply MuS, and we will include a model-only formulation in the Appendix.
>
> ### Questions: Impact of MuS and relation of stability metric with respect to the original classifier.
>
> We do not provide any guarantees for the original classifier. Our guarantees are only with respect to the smoothed classifier. Because the original classifier may be an arbitrary $h : \mathbb{R}^n \to [0,1]^m$ function, it is challenging to establish any kind of formal guarantees. Only by smoothing the original ("base") classifier can we establish any kind of non-trivial guarantees. However, it would be interesting to consider what additional properties we could guarantee if one could assume certain qualities about the original classifier.

---

> > ### Comment · Reviewer_JJJj · 2023-08-11
> > **Response**
> >
> > I thank the authors for the detailed response. I will maintain my "weak accept" review.

---

### Official Review · Reviewer_dPnc · 2023-07-12

**Soundness:** 4 excellent
**Presentation:** 3 good
**Contribution:** 3 good
**Rating:** 7
**Confidence:** 4

**Summary:**

This paper presents a technique for extracting feature attributions that are certifiably stable in the sense that the model's predictions are consistent on supersets of attributed features. As the title suggests, the approach is based on multiplicative smoothing, a novel type of Bernoulli smoothing based on masking, rather than perturbing, input features. Notably, the paper shows how to construct a distribution with coordinate-wise Bernoulli features that requires significantly fewer samples to obtain a certificate.

**Strengths:**

This work introduces a new type of attribution stability that is not addressed by prior work on robust feature attribution, and presents rigorous yet practical techniques for obtaining stable explanations.

As the core technique is based on randomized smoothing, it is applicable broadly to different types of models and even different feature attribution methods. Thus, it complements a large body of existing work, and gives concrete, practical ways to improve it with respect to stability.

While smoothing-based certification methods are typically very expensive, the structured dependency sampling described in section 3.3 gives a clever way to avoid this, making the approach reasonably inexpensive.

The empirical results give a sense of what the guarantees look like on real data, and compare several widely-used attribution methods under the certification regime.

The writing is polished and clear on most of the important points. Overall the paper is an enjoyable read.

**Weaknesses:**

The accuracy penalty imposed by multiplicative smoothing is large (a related small point: the horizontal axis between figures 4 and 5 isn't the same, making these results a little harder to line up). It doesn't seem that the models were trained with augmentations that match smoothing noise, or that denoising was used on smoothing samples, so there may be opportunities to improve on this. Without improvements, the results in Figure 5 pull somewhat against the practical significance of the work.

The writing is a bit unclear around the sufficiency of Lipschitzness and masking equivalence. A less-careful read of the paper that focuses more on early sections might come away with the impression that most techniques for obtaining Lipschitz models will suffice for the stability guarantees sketched in remark 2.6. The authors might consider moving some of the intuitions from proposition 3.1 into this discussion.

The notion of stability at the center of this work is different than what I would have expected, being familiar with some prior work on this topic and reading the abstract. I agree that the model needs to play a central role in characterizing the stability of explanations, but it's less clear that being stable to added features is always an important property. The paper might have wider appeal if the authors are able to justify this, using examples or otherwise (Figure 1 is indeed an example, but doesn't address the question of why the classifier's behavior over the right two images is harmful or undesirable).


**Questions:**

1. Were the models used in figures 4 and 5 trained on appropriately augmented data?
2. Have you attempted to use denoising methods as an intermediate step after sampling from the smoothing distribution?
3. Are there other existing methods for obtaining Lipschitz models, perhaps without smoothing, that could also work with your certificates?

**Limitations:**

The paper does not have an explicit limitations section, but these points are discussed where it is appropriate.

---

> ### Author Rebuttal · Authors · 2023-08-10
>
> We thank the Reviewer for their positive impression of our paper and helpful comments on how to improve readability, especially on the notion of masking equivalence and our particular definition of stability. Moreover, since our submission, we have fine-tuned our models for longer and rerun a number of experiments, generally yielding substantially better numerics, especially on ResNet50. We include some examples in the Rebuttal Supplementals, and we respond to each of the Reviewer's comments below.
>
> ### Weakness 1 / Question 1: Accuracy penalty and training with augmented data.
>
> After our submission, we have had the opportunity to rerun some of our evaluations under more generous time constraints. In particular, the original numbers for Figures 4 and 5 were run with models after only one epoch of fine-tuning on the properly augmented data. Our rerun experiments use models with five epochs of fine-tuning, and we achieve better results, substantially so for ResNet50. We have included some of the new figures in Section 2 of the Rebuttal Supplementals.
>
> In addition, we have reworked the discussion to better emphasize that the case of $\lambda = 1/16$ is essentially a stress test since this tends to be more noise than what many smoothing papers employ in their experiments. Notably, because smoothed variants of Vision Transformer tend to outperform the smoothed ResNet50 counterpart, we will include additional discussion of how these experiments can inform model selection for potential users of MuS.
>
> ### Weakness 2: Exposition on the sufficiency of Lipschitzness and masking equivalence.
>
> This is a great suggestion on where to position the presentation on masking equivalence. We will work this into our revised manuscript.
>
> ### Weakness 3: "Non-standard" notion of stability and its desirability.
>
> The Reviewer makes an acute observation that the naming of "stability" is tricky. Indeed, we considered other possible names (e.g. "monotone", "sufficient", "convincing", "upwards-consistent", "robust", etc) but each choice came with its own advantages and disadvantages. In the end, we felt that "stable" was the most descriptive name that was the least potentially confusing, even though it risks conflation with terminology from adversarial robustness. We will revise the exposition to better distinguish and clarify our definitions.
>
> We also thank the Reviewer for pointing out that the benefits of stability are not sufficiently well-motivated. We plan to better motivate the need for formal guarantees by using more examples from medicine. In particular, we have constructed an example using chest X-ray images (see Section 1 of Rebuttal Supplementals) to use as a non-trivial example in our Introduction. We believe that this should better motivate readers outside of the ML-explainablity community.
>
> ### Question 2: Denoising steps.
>
> We have not attempted to use denoising methods as an intermediate step. This is a great suggestion and seems like a viable strategy for improving performance. Importantly, however, directly applying additive $L^p$-based denoising methods would not work in our setting, as our multiplicative noise drops features. An appropriate "denoiser" in our noising scheme therefore needs to fill in the dropped features via some data imputation method. Investigating such a denoiser is an interesting future direction.
>
> ### Question 3: Other methods for obtaining Lipschitz models.
>
> We are not aware of other methods for obtaining Lipschitz models that conform to our criteria for stability. To our knowledge, our method is the only one among smoothing-based methods. We are not aware of any non-smoothing approaches, but this would be an interesting direction for future research.

---

> > ### Comment · Reviewer_dPnc · 2023-08-16
> > **Reply to rebuttal**
> >
> > Thank you for taking the time to write this detailed rebuttal. Your suggestions for addressing the weaknesses mentioned in my original review look reasonable. A few comments on your answers to my questions below.
> >
> > * [Q1] I'm glad to see that training with augmentation improves things!
> > * [Q2] Certainly, additive denoising isn't right. Something based on a diffusion model, along the lines of [1], could be promising.
> > * [Q3] Agreed, and to be clear, I believe that this paper stands on its own either way.
> >
> > I'll maintain my score and support for this paper.
> >
> > [1] Nicholas Carlini, Florian Tramer, Krishnamurthy Dj Dvijotham, Leslie Rice, Mingjie Sun, J. Zico Kolter. (Certified!!) Adversarial Robustness for Free! https://arxiv.org/abs/2206.10550

---

### Official Review · Reviewer_7LaM · 2023-07-23

**Soundness:** 4 excellent
**Presentation:** 3 good
**Contribution:** 3 good
**Rating:** 7
**Confidence:** 2

**Summary:**

The draft addresses an important problem of feature attribution method selection and formalizes a notion of attribution stability to do this. The approach is based on modifying classifiers to make them Lipschitz wrt to feature masking. Once done, the new classifier has provable radii of stability defined as L1 distances within which the found sparse explanation is sufficient and adding new features will not change the prediction.

The approach is tested on vision and NLP tasks with four different explanation methods.

Despite some weaknesses the paper is a good contribution to the field of feature attribution explanations.

**Strengths:**

- new measurable definition of explanation stability
- a method to convert any classier into a one that permits the stability calculation
- theoretical analysis; efficient computation
- comprehensive empirical evaluation

**Weaknesses:**

- The need to modify the classifier, before guarantees can be calculated, somewhat diminishes the practical value. In practice, we would be interested in the classifier that actually does the prediction, not a smoothed version of it; as the authors in Sec. 4.1 measured the drop of accuracy from the smoothing to be tens of %s, it could be a no-go for critical classifiers.
- The proposed stability is operationalized only for binarized explanations and features as image patches, which throws the classifier into the out-of-distribution regime and, on top of smoothing, further complicates drawing conclusions from the approach. In fact all feature attribution methods compared with proposed stability are continuous and require binarization before they can be used.


**Questions:**

- typo in the last word of line 44
- nitpick: logits are usually defined as pre-softmax values, not probabilities (line 71)
- Figures are not readable in black and white


**Limitations:**

No dedicated Limitations section but limitations mentioned in the text

---

> ### Author Rebuttal · Authors · 2023-08-10
>
> We thank the Reviewer for their positive reception of our paper. We have rerun some experiments under more generous time budgets to improve our experimental results and better address the highlighted weaknesses. We include some of these in the Rebuttal Supplementals, and we address each of the Reviewer's comments below.
>
> ### Weakness 1: Need to modify the classifier and weakness of our Figure 5 evaluations.
>
> We thank the Reviewer for highlighting a need to better discuss this weakness of smoothing methods, especially in the context of our initial experiments in Figure 5. After our submission, we reran a number of our experiments after fine-tuning our models for more epochs and observed substantial performance gains, for instance on experiments of Figure 5 (see Section 2 of Rebuttal Supplementals).
>
> Although the new numbers are improved, they do still exhibit similar trends, and we comment on both the old and new numbers here. The Reviewer is correct in pointing out that under greater noise (i.e. $\lambda = 1/16$) there is non-trivial accuracy degradation. Fortunately, these are much improved in our new experiments. We will revise our discussion to also emphasize that the extreme-noise case is effectively a stress test, as $\lambda = 1/16$ is more noise than what many smoothing papers consider.
>
> Moreover, we will also include commentary on the fact that one unit of radius in our setting can certify a much greater amount of perturbation to the input than what classical additive smoothing methods can achieve. For instance, we segment 3x224x224 ($n=150528$ features) images into 64 patches, so one unit of radius is then 1/64 of the image. By contrast, many additive smoothing methods struggle to certify more than tens out of the 150k+ features. This difference is largely due to the fact that we are Lipschitz with respect to input feature maskings (which we express as $L^1$-Lipschitz to the $\alpha$), rather than $L^p$-norm Lipschitz to the input.
>
> In addition, we have expanded our discussion to state that the experiments of Figure 5 may be used to guide model selection for potential users of MuS. In particular, at least on vision-based problems, Vision Transformer experiences higher smoothed accuracy compared to ResNet50. This shows that transformer-based architectures may be more compatible with MuS than CNN-based models.
>
> ### Weakness 2: Focus on binarized explanations.
>
> The objective of our work is to achieve provable guarantees for feature attributions. To do so, we first focus on the simpler case of binary attributions. Importantly, explanation-relevant guarantees currently do not exist for either continuous-valued or binary attributions. The Reviewer is correct in identifying that the continuous-valued case is more general, however, it is not trivial as to (1) what kind of formal properties should be imposed on continuous-valued attributions, and (2) how one may achieve such properties in a computationally reasonable manner. For instance, it is not clear to us how one should specify properties like stability for continuous-valued attributions. We thus focus on the binary case first.
>
> ### Questions: Typos, wording, etc.
>
> We thank the Reviewer for pointing these out. We have corrected the typo on L44, as well as others that we have found. We have also reworded "logit" to "class probability" and "confidence" where applicable, and we believe that this change also makes the paper more accessible to a non-ML audience while improving its technical clarity. Moreover, we will modify our figures to improve black/white readability where possible. We show in Section 2 of the Rebuttal Supplementals of one such example with different line styles.

---

> > ### Comment · Reviewer_7LaM · 2023-08-10
> > **Thank you**
> >
> > Thanks for the detailed reply - keeping the Accept note.
> >
> > (review slightly edited for clarity/typos)

---

### Official Review · Reviewer_Qhau · 2023-07-25

**Soundness:** 3 good
**Presentation:** 3 good
**Contribution:** 2 fair
**Rating:** 5
**Confidence:** 3

**Summary:**

This paper studies the stability of binary attributions. The attribution is defined to be stable if the prediction does not change when adding additional features. The multiplicative smoothing is proposed to achieve the Lipschitz condition, which is proved to infer the relaxation of stability. Experiments verify that the smoothing technique is useful for stability.

**Strengths:**

1. The paper is well-written and easy to understand. The problem of attribution stability is novel.
2. The proposed multiplicative smoothing is effective under the defined stability setting and can be applied to any models.

**Weaknesses:**

1. For the smoothing technique, my main concern is its application scope, which currently focuses only on binary attribution methods. Although a method is proposed to convert non-binary attributions into binary ones, this conversion process disregards the continuous information in the attribution maps. The potential loss of information during this conversion is significant, especially when dealing with features grouped into 64 patches.
2. The motivation behind this work can be emphasized further. While it is mentioned that explanations can be fragile, more details are necessary to inform readers about the specific circumstances under which explanations may fail. In order for the stabilty of explanations to be meaningful, classifiers must rely solely on attributions, or more features added on attributions, to make correct decisions. However, it remains unclear why practitioners would opt to use attributions for decision-making when they already have access to the original input data.
3. The accuracy drop can also be a weakness of the proposed method. While the decrease is expected, a significant decrease in accuracy may suggest a limited usefulness of the method.

**Questions:**

1. I'm interested in the performance comparison of additive smoothing and multiplicative smoothing on the stability setting.

**Limitations:**

See weakness

---

> ### Author Rebuttal · Authors · 2023-08-10
>
> We thank the Reviewer for their constructive comments. We will include additional exposition, discussion, and examples in our revised manuscript to better motivate our work and clarify our contributions. Furthermore, we have attached a Rebuttal Supplemental document with relevant material.
>
> ### Weakness 1: Application scope of binary attributions.
>
> The Reviewer accurately notes that the continuous-valued case is indeed more general. However, it is also more challenging: it is not clear how to formally specify many explanation-relevant properties, nor how one may achieve such properties. For instance, it is not clear to us how stability-like properties should be formulated for the continuous case. Surprisingly, formal explanation properties of any kind are rarely studied, even in the binary case. We therefore begin with binary attributions as a first step towards explanation stability.
>
> Moreover, we do not aim to introduce novel binary attributions. Rather, our simple binarization schemes are solely intended to show a proof-of-concept that one can certify stability guarantees via existing methods. Importantly, to our knowledge, there do not exist any continuous nor binary attribution methods that explicitly consider our notion of stability. We therefore test MuS with very popular methods (e.g. LIME, SHAP, etc), which all happen to be continuous-valued and are far more well-known than any binary variant.
>
> ### Weakness 2a: Classifier must solely rely on attributions.
>
> The Reviewer's observation is correct. In fact, this highlights an important question: how does one evaluate the "quality" of a feature selection? Our choice for quality evaluation is most natural in the context of vision models, wherein revealing parts of an image is analogous to revealing information of the input. If another quality metric were used (e.g. Reference [15]), then our stability definition will no longer apply.
>
> ### Weakness 2b: Usefulness of feature attributions.
>
> A major ongoing use case of feature attributions is in the application of machine learning to medicine [1,2,3]. Many such explanation methods are already deployed, but there is often a glaring absence of formal guarantees. This is especially concerning, because such explanations are intended to help medical personnel make critical decisions about patient care. As we cannot expect a general user to understand the technical nuances of ML explanations, it is important that these explanations should be engineered with useful formal guarantees by construction. Our work on explanation stability is a step in this direction.
>
> We will also include a medical example from Section 1 of the Rebuttal Supplementals. Here, a detector from TorchXRayVision predicts an image to have pneumonia with 25.5% confidence. Suppose a doctor then observes two similar feature selections with dramatically different pneumonia detection confidences (e.g. 29.56% vs 50.05%), they may lose trust in the model and explanation. The objective of our work is to analyze and understand how to prevent such scenarios.
>
> ### Weakness 3: Accuracy drop of smoothing.
>
> Since our submission, we have fine-tuned the models for more epochs and obtained better evaluation numerics. An example of one such new improvement is in Section 2 of the Rebuttal Supplementals, in which we rerun the Figure 5 experiments and observe substantial accuracy improvements, especially for ResNet50.
>
> We will also emphasize that these numbers include heavy noise stress tests (i.e. $\lambda = 1/16$). This is a higher noise than what is typical in smoothing papers and leads to very small $1/16$-Lipschitz constants.
>
> Moreover, we will include discussion that the relative difference in smoothed classifier accuracy of these experiments are useful in guiding model selection for MuS users. In particular, this experiment demonstrates the relative advantage of transformer-based architectures over CNN-based ones on vision tasks.
>
> ### Question 1: Comparison with additive smoothing.
>
> We thank the Reviewer for suggesting this experiment. Note that classic additive smoothing approaches, which add noise directly to the input, use a radius that is typically too small to cover the deletion of even one feature, so we would not expect it to work well. We verified this in the following experiment similar to Section 4.1, with $L^1$-smoothing noise from distributions as in Reference [21, 27]:
>
> We use SHAP-top25% and test how often Vision Transformer is incrementally stable with radius $r \geq 1$ on $\lambda =  8/8, 4/8, 2/8, 1/8$. We use $N=250$ samples from ImageNet. The numbers are as follows, and are also plotted in Section 3 of the Rebuttal Supplementals:
>
> Add: 0.184, 0.320, 0.360, 0.156
>
> MuS: 0.716, 0.660, 0.512, 0.364
>
> This shows that MuS consistently outperforms classic additive smoothing in terms of achieving radius $r \geq 1$ incremental stability, as expected. We will include this experiment and discussion in our revised manuscript.
>
> We note that the Reviewer may be also interested in using additive smoothing not in the classical setting, but to directly smooth the $\alpha$ parameter in the explainable model framework from our paper. However, obtaining stability in this way is not theoretically possible, which is formalized in Proposition 3.1. This is precisely why we had to develop multiplicative smoothing.
>
> ### Supplemental References:
>
> [1] A. M. Antoniadi, Y. Du, Y. Guendouz, L. Wei, C. Mazo, B. A. Becker, and C. Mooney. Current Challenges and Future Opportunities for XAI in Machine Learning-Based Clinical Decision Support Systems: A Systematic Review. Applied Sciences, 2021.
>
> [2] W. T. Hrinivich, T. Wang, and C. Wang. Interpretable and explainable machine learning models in oncology. Frontiers in Oncology, 2023.
>
> [3] S. Khedkar, P. Gandi, G. Shinde, and V. Subramanian. Deep Learning and Explainable AI in Healthcare Using EHR. Deep Learning Techniques for Biomedical and Health Informatics, 2020.

---

> > ### Comment · Reviewer_Qhau · 2023-08-15
> >
> > Thanks for the author's detailed response, which has addressed most of my concerns. The work's contribution is commendable, especially considering it as a first attempt in explaining stability. Therefore, I will increase my score to 5.

---

### Author Rebuttal · Authors · 2023-08-10

We thank the Reviewers for their time and constructive feedback. The Reviewers have raised many insightful comments and useful suggestions on how to improve the expositional narrative, technical presentation, and experimental evaluations. In addition, the Reviewers have also suggested a number of additional examples and experiments that we believe will help bolster the presentation of our paper. We have attached a Rebuttal Supplemental material that includes a selection of the relevant example and experiment figures.

---

### Decision · Program_Chairs · 2023-09-21

**Decision:**

Accept (poster)

**Comment:**

This paper looks at stability of model predictions around a set of explanatory features, giving a formal notion of a radius of stability which can be achieved if the model is λ-Lipschitz (i.e. the value does not change too fast). They propose a smoothing method, multiplicative smoothing or MuS, which can be applied to any classifier by sampling masks at inference time, and measure the quality of stability guarantees, stability-accuracy tradeoff, and stability around explanations from different methods such as SHAP or gradients.

Reviewers agree that this is a strong paper, appreciating elegant and effective solution to the important problem of explanation stability, as well as the theoretical analysis paired with clear empirical evaluation. There were some concerns about the limitations of the binary-masking formulation, evaluation cost, as well as the accuracy drop caused by the smoothing, and also that the description of Lipschitz-smoothness could be hard to follow - although many of these points were addressed by the rebuttal and two reviewers raised their scores as a result of the discussion.